

# Numerical strategies for representing Richards' equation and its couplings in snowpack models

Kévin Fourteau[1], Julien Brondex[1], Clément Cancès[2], and Marie Dumont[1]

[1]Univ. Grenoble Alpes, Université de Toulouse, Météo-France, CNRS, CNRM, Centre d'Études de la Neige, Grenoble, France
[2]Univ. Lille, CNRS, Inria, UMR 8524 - Laboratoire Paul Painlevé, 59000, Lille, France

**Correspondence:** Marie Dumont (marie.dumont@meteo.fr)

**Abstract.** The physical processes of heat conduction, liquid water percolation, and phase changes govern the transfer of mass and energy in snow. They are therefore at the heart of any physics-based snowpack model. In the last decade, the use of Richards' equation has been proposed to better represent liquid water percolation in snow. While this approach allows the explicit representation of capillary effects, it can also be problematic as it usually presents a large increase in numerical

complexity and cost. This notably arises from the problem of applying a water retention curve in a fully-dry medium such as snow, leading to a divergence and degeneracy in Richards' equation. Moreover, the difficulty of representing both dry and wet snow in a single framework hinders the concomitant solving of heat conduction, phase changes, and liquid percolation. Rather, current models employ a sequential approach, which can be subject to non-physical overshoots. To treat these problems, we propose the use of a regularized water retention curve (WRC), that can be applied to dry snow. Combined with a variable

switch technique, this opens the possibility of solving the energy and mass budgets in a fully consistent and tightly coupled manner, whether the snowpack contains dry and/or wet regions. To assess the behavior of the proposed scheme, we compare it to other implementations based on loose-coupling between processes and on the state-of-the-art strategies in snowpack models. Results show that the use of a regularized WRC with a variable switch greatly improve the robustness of the numerical implementation, consistently allowing timesteps greater or equal to $900\,\text{s}$, which results in faster and cheaper simulations.

Furthermore, the possibility of solving the physical process in a fully-coupled and concomitant manner results in a slightly reduced error level compared to implementations based on the traditional sequential treatment. However, we did not observe any numerical oscillations and/or divergence sometimes associated with a sequential treatment. This indicates that a sequential treatment remains a potentially interesting tradeoff, favoring computational cost for a small decrease in precision.

## 1  Introduction

The wetting of snowpacks is a crucial stage in their evolution over time, with direct impacts on their environment. Notably, the intensity and timing of melt runoff have strong implications for the water availability in hydrological basins (Hock et al., 2019; Barnhart et al., 2020). Likely, the quantification of wet snow avalanche hazard relies on a precise determination of the liquid water content in snowpacks and of the depth of the percolation front (Baggi and Schweizer, 2009; Eckert et al., 2024). Liquid water percolation, also plays a crucial role in the formation of ice lenses and crusts (Wever et al., 2016; Quéno et al.,



2020), with direct implications for the local ecosystems (Tyler, 2010; Hansen et al., 2010). In this context, numerical snowpack models play a key role to assess the melting and wetting of snowpacks in various locations and under various conditions.

Historically, liquid water percolation in snowpack models has been implemented through the use of a so-called bucket-scheme, where snow layers are expected to retain liquid water until a certain threshold, after which all liquid water is instantaneously transferred downward (Bartelt and Lehning, 2002; Vionnet et al., 2012; Sauter et al., 2020). While this implementation

is numerically efficient, it cannot capture certain effects, such as capillary barriers or capillary rise. To foster a more accurate representation of liquid water percolation, it has been proposed in the last decade to model it based on Richards' equation (Richards, 1931; Wever et al., 2014; D'Amboise et al., 2017). This more advanced description has notably been shown to better capture the timing associated with the wetting of the snowpack (Wever et al., 2015), although it does not address the challenge of simulating preferential flow, which is crucial to fully capture the complexity of liquid water percolation in snowpacks.(Wever

et al., 2016; Moure et al., 2023).

In this framework, the simple bucket-scheme is replaced by a partial differential equation of liquid water mass budget with capillary effects. However, Richards' equation can be notoriously difficult to solve numerically, due to its non-linearities and potential degeneracy (Farthing and Ogden, 2017). Notably, it is usually associated with an adaptive time stepping strategy (e.g. Celia et al., 1990; Forsyth et al., 1995; Wever et al., 2014; D'Amboise et al., 2017), as the use of iterative methods to solve

non-linear equations can fail at large timesteps. Therefore, the implementation of Richards' equation in snowpack models can imply a significant increase in numerical cost, hindering its broad use, in particular for simulations over large areas and long periods. For instance, the current implementation of Richards' equation in the detailed snowpack model Crocus (D'Amboise et al., 2017) is known to display signs of numerical instabilities and to require small timesteps of the order of 30 seconds or less (D'Amboise et al., 2017, and M. Lafaysse, personal communication, 2024), while the model is usually run with a 900 s

timestep. This drastically hinder the routine use of Richards' equation in Crocus, as the two-order of magnitude increase in numerical cost can be considered as a too expensive trade-off. Moreover, due to the difficulty of representing both dry and wet snow in a unified framework for Richards' equation, current implementations in snowpack models assume that a small amount of liquid water is always present and need to modify the capillary behavior of snow in consequence (Wever et al., 2014; D'Amboise et al., 2017). This lack of unified treatment of dry and wet snow hinders the concomitant numerical solving

of liquid water percolation with other physical processes (such as phase change or heat conduction). Rather, Richards' equation is solved sequentially (Wever et al., 2014; D'Amboise et al., 2017).

The main goal of this article is to investigate how the specific numerical implementation of liquid water percolation in snow affects the behavior of snowpack models, with a focus on their robustness and numerical cost. For this, we rely on various techniques proposed in applied mathematics, where the efficient solving of Richards' equation has been an active research

subject (e.g. Forsyth et al., 1995; Sadegh Zadeh, 2011; Farthing and Ogden, 2017; Bassetto et al., 2020). Specifically, we explore whether the use of regularized capillary laws and the concomitant solving of Richards' equation with other physical processes are beneficial for the numerical behavior of snowpack models. The article is organized as follows. Section 2 presents a consistent system of equations describing energy and mass conservation in snowpacks that applies naturally to both dry and wet snow. Section 3 presents toy-models based on different numerical implementations of the heat and mass budget equations





and on the standard implementations used in state-of-the art detailed snowpack models as well. Simple test cases, representing three distinct situations of liquid water infiltration in snowpacks are presented in Section 4. Finally, the performance of the toy-models on these test cases are discussed in Section 5.

## 2 Deriving a consistent system of equations for the energy and mass budgets in snowpacks

While a real snowpack is a complex 3D structure composed of intertwined ice, air, and liquid water, such complexity is chal-
lenging to model. Rather, snowpack models rely on a macroscopic and 1D framework (e.g. Jordan, 1991; Bartelt and Lehning, 2002; Vionnet et al., 2012; Sauter et al., 2020). Here, macroscopic means that snow is not treated as an actual multiphasic medium, but rather as a homogeneous material (Torquato, 2002). In this framework, snow is characterized by macroscopic (sometimes referred to as effective) material properties, such as its thermal capacity (Calonne et al., 2014) or its hydraulic conductivity (Calonne et al., 2012).
The core of any snowpack model thus requires to determine and solve the equations governing the evolution of the macroscopic energy and mass contents of snow. This can be done from the first principles of energy and mass conservation, complemented by material laws characterizing the effective material properties. The goal of this section is to present a set of equations that govern the physical evolution of a snowpack and that apply both in dry and wet snow. Note that while this article assumes a 1D framework, as usually done in snowpack models, it could easily be transposed to a 2D or 3D configuration.


### 2.1 Snow internal energy conservation

As a first equation governing the evolution of a snowpack, we consider the energy conservation of snow, understood here as the combination of the ice matrix and the air within (and excluding potential liquid water). In the case of dry snow, and neglecting the influence of water vapor for simplicity, the temporal evolution of the snow energy is given by a classical conservation
equation, i.e.:

$$\partial_t h_{\mathrm{s}} + \nabla \cdot F_{\mathrm{cond}} = Q_{\mathrm{abs}} \qquad (1)$$

where $h_{\mathrm{s}}$ is the energy content of snow (expressed in $\mathrm{J\,m^{-3}}$), $F_{\mathrm{cond}}$ the heat conduction flux (in $\mathrm{W\,m^{-2}}$), and $Q_{\mathrm{abs}}$ a volumetric energy source (in $\mathrm{W\,m^{-3}}$), for instance the energy source due to shortwave absorption within the snowpack (van Dalum et al., 2019; Picard and Libois, 2024). Moreover, we classically assume that the heat conduction flux $F_{\mathrm{cond}}$ follows Fourier's law:

$$F_{\mathrm{cond}} = -\lambda \nabla T \qquad (2)$$

where $\lambda$ is the thermal conductivity of snow (in $\mathrm{W\,K^{-1}\,m^{-1}}$) and $T$ its temperature (K). These equations, and the value of the snow thermal conductivity in terms of the microstructure, can for instance be derived from homogenization methods (e.g.



Boutin et al., 2010; Calonne et al., 2011, 2014; Bouvet et al., 2024). Moreover, the temperature and the energy content are related through the thermal capacity of snow:

$$h_{\mathrm{s}} = \int_{T_0}^{T} c_{\mathrm{p}} dT = c_{\mathrm{p}}(T - T_0) \tag{3}$$

where $T_0$ is a reference temperature, and $c_{\mathrm{p}}$ is the volumetric thermal capacity of snow (in $\mathrm{J\,m^{-3}\,K^{-1}}$; Tubini et al., 2021). Note that we have assumed for simplicity that $c_{\mathrm{p}}$ does not depend on temperature, as regularly done in snowpack models (e.g. in the Crocus model; Vionnet et al., 2012). Also, we made the choice to take the fusion temperature of pure water as the reference temperature. This implies that the energy of snow is strictly negative below its fusion point, and vanishes at the fusion

temperature. Using homogenization methods, the volumetric thermal capacity of snow can be shown (e.g. Calonne et al., 2014) to be

$$c_{\mathrm{p}} = \phi_{\mathrm{i}} c_{\mathrm{i}} + (1 - \phi_{\mathrm{i}}) c_{\mathrm{a}} \simeq \phi_{\mathrm{i}} c_{\mathrm{i}} \tag{4}$$

where $\phi_{\mathrm{i}}$ is the ice volumetric fraction, and $c_{\mathrm{i}}$ and $c_{\mathrm{a}}$ the volumetric thermal capacity of ice and air, respectively.

Specificities arise in the case of wet snow. Indeed, due to the presence of latent heat during the phase transition between ice and liquid water, the temperature of snow is blocked at the fusion temperature in case of ice-liquid water coexistence. The wet part of a snowpack thus becomes isothermal, and the internal thermal gradient vanishes, leading to a zero heat conduction flux. This phenomenon is already taken naturally into account in Eq 2, and thus does not require further development. A second point is that in the case of wet snow, input or removal of energy leads to phase change rather than temperature change. This

phenomenon can be taken into account in Eq 1 by introducing an energy source term, related to the melting and refreezing of water (e.g. Tubini et al., 2021). The energy budget of ice matrix in snow is thus governed by:

$$\partial_t h_{\mathrm{s}} - \nabla \cdot (\lambda \nabla T) = Q_{\mathrm{abs}} + Q_{\mathrm{freeze}} \tag{5}$$

where $Q_{\mathrm{freeze}}$ (in $\mathrm{W\,m^{-3}}$) is the rate of latent heat released/absorbed during the freezing/melting of water (taken positive for freezing).

## 2.2 Liquid water conservation

Since the heat budget and temperature of snow are directly related to the melting and freezing of water, it is necessary to also treat the liquid water budget. This can be done with the use of Richards' equation (Wever et al., 2014, 2015, 2016; D'Amboise et al., 2017). Note that in this paper we only consider "matric" water flow, and do not consider fast preferential flow (Vogel et al., 2000; Lewandowska et al., 2004; Wever et al., 2016). This limitation is discussed in Section 5.4. Under these conditions,

the mass conservation of liquid water is given by:





$$\rho_{\mathrm{w}} \partial_t \theta_{\mathrm{w}} + \rho_{\mathrm{w}} \nabla \cdot F_{\mathrm{w}} = -M_{\mathrm{freeze}} \tag{6}$$

where $\theta_{\mathrm{w}}$ is the volumetric liquid water content (LWC; expressed in $\mathrm{m}^3$ of water per $\mathrm{m}^3$ of snow), $\rho_{\mathrm{w}}$ the density of liquid water (assumed constant; in $\mathrm{kg\,m}^{-3}$), $F_{\mathrm{w}}$ is the liquid water flux (in $\mathrm{m}^3$ of water per $\mathrm{m}^2$ of snow per s), and $M_{\mathrm{freeze}}$ is the rate of freezing water (counted positive in the case of freezing and expressed in kg of water per $\mathrm{m}^3$ of snow per s). The rate of

freezing water $M_{\mathrm{freeze}}$ is directly related to the rate of energy release/absorption $Q_{\mathrm{freeze}}$ of Eq 5 through $Q_{\mathrm{freeze}} = L_{\mathrm{fus}} M_{\mathrm{freeze}}$, with $L_{\mathrm{fus}}$ the specific enthalpy of fusion of water (in $\mathrm{J\,kg}^{-1}$).

The liquid water flux is assumed to follow the Darcy-Buckingham law (Sposito, 1978):

$$F_{\mathrm{w}} = -K^{\mathrm{sat}} k_{\mathrm{r}} (\nabla \psi + \cos(\gamma)) \tag{7}$$

where $K^{\mathrm{sat}}$ is the saturated hydraulic conductivity (in $\mathrm{m\,s}^{-1}$), which does not depend on the LWC but only the properties of the ice matrix, $k_{\mathrm{r}}$ the relative hydraulic conductivity, that depends on the LWC, $\psi$ the liquid matric potential (in $m$ and that can be negative due to capillary forces), and $\gamma$ the slope angle. In the same way that the snow energy and temperature are related through Eq 3, the LWC and the matric potential are related through the use of a water retention curve (WRC). This WRC can be expressed with the use of, for instance, the van Genuchten or Brook models (Brooks and Corey, 1964; van Genuchten, 1980;

Lenhard et al., 1989). In this article, and consistently with previous snow-related work (Wever et al., 2014; D'Amboise et al., 2017), we use a van Genuchten model for the WRC in snow:

$$\theta_{\mathrm{w}} = \begin{cases} \theta_{\mathrm{r}} + (\theta_{\mathrm{s}} - \theta_{\mathrm{r}})(1 + |\alpha \psi|^n)^{\frac{1-n}{n}} & \text{if } \psi < 0 \\ \theta_{\mathrm{s}} & \text{otherwise} \end{cases} \tag{8}$$

where $n$ and $\alpha$ (in $\mathrm{m}^{-1}$) are parameters characterizing the shape of the WRC and which depends on the snow microstructure (Yamaguchi et al., 2012), and $\theta_{\mathrm{r}}$ and $\theta_{\mathrm{s}}$ are referred to as the residual and saturation LWC, respectively. Physically, $\theta_{\mathrm{s}}$ cor-

responds to the maximum amount of liquid water a snow sample can hold (thus typically assumed to correspond to the total porosity), and $\theta_{\mathrm{r}}$ to the minimum amount of water that can be reached with draining through capillary and gravity forces.

To solve Richards' equation, it is as well necessary to provide an expression for the dependence of the relative hydraulic conductivity $k_{\mathrm{r}}$ to the liquid water content. Following Mualem (1976), in complement of the van Genuchten WRC, the hydraulic

conductivity is usually taken as

$$k_{\mathrm{r}} = \sqrt{S} \left( \frac{\int_0^S \frac{dS}{\psi}}{\int_0^1 \frac{dS}{\psi}} \right)^2 = \sqrt{S} \left( 1 - \left( 1 - S^{\frac{n}{n-1}} \right)^{\frac{n-1}{n}} \right)^2 \tag{9}$$



where $S = \frac{\theta_\mathrm{w} - \theta_\mathrm{r}}{\theta_\mathrm{s} - \theta_\mathrm{r}}$ is referred to as the saturation degree. It can be noted that the relative hydraulic conductivity is an increasing function of $\theta_\mathrm{w}$, that vanishes at the residual LWC $\theta_\mathrm{r}$ and reaches unity value at water saturation.

However, Richard's equation presents specific problems in the case of water-saturated and fully-dry materials. In the case of a water-saturated materials, the liquid water content reaches a maximum value, while the matric potential can keep increasing. In other words, in the saturated case, the equation becomes degenerate and can no longer be expressed in terms of liquid water content. This is a classical difficulty associated with Richard's equation, usually circumvented by using $\psi$ as the primary variable to describe the material (Celia et al., 1990; Farthing and Ogden, 2017). It can also be treated using a variable switch

technique, effectively changing from $\theta_\mathrm{w}$ to $\psi$ as a primary variable depending on the degree of saturation of the material (Diersch and Perrochet, 1999; Krabbenhøft, 2007; Bassetto et al., 2022). This latter technique is explored below in the article. The problem of a fully-dry material is more specific to snow, and requires a dedicated treatment.

### 2.2.1 The problem of Richards's equation for dry snow

Indeed, in their usual forms, the WRCs used as material laws in Richards' equation do not apply in dry snow. In Eq. 8, the matric potential $\psi$ diverges when the LWC approaches its residual value $\theta_\mathrm{r}$. In material solely driven by capillary and gravity flow, this implies that $\theta_\mathrm{w}$ cannot drop below this residual value, i.e. that liquid water is always present. To the contrary, snow can become fully dry when energy is removed and the residual liquid water freezes. In such a case, the LWC can fall below the residual liquid water content and even vanish. Then, the matric potential $\psi$ becomes undefined. In snowpack models, this issue

of dry snow has been circumvented in previous implementations of Richards' equation by keeping a small liquid water content, even in the case of snow below its fusion temperature (Wever et al., 2014; D'Amboise et al., 2017). This however hinders a consistent treatment of coupled heat and liquid water budget, as snow below the fusion is considered dry while solving the heat budget but wet while solving the liquid water budget. Furthermore, this technique requires the residual LWC to be artificially modified, in order to remain strictly below the liquid water content at all times. Thus, snow containing very little liquid water

will tend to percolate, even tough the unmodified WRC would rather imply that the liquid water should be hold still under capillary forces.

This issue of the disappearance of a phase and of the divergence of the capillary forces is not only encountered in snow modeling. It is for instance present in underground nuclear-waster storage, where phases can appear and disappear (e.g. Bourgeat et al., 2009). To the best of our knowledge, two types of solutions to this problem have been proposed. First, the gradient of ma-

tric potential can be rewritten in terms of the gradient of liquid water content using the chain-rule (similarly to Bourgeat et al., 2009). The liquid water flux is thus taken as $-K^\mathrm{sat} k_\mathrm{r} \left( \nabla \psi + \cos(\gamma) \right) = -K^\mathrm{sat} k_\mathrm{r} \left( \partial_{\theta_\mathrm{w}} \psi \nabla \theta_\mathrm{w} + \cos(\gamma) \right)$. While the derivative $\partial_{\theta_\mathrm{w}} \psi$ diverges near the residual point, the product $-K^\mathrm{sat} k_\mathrm{r} \partial_{\theta_\mathrm{w}} \psi$ vanishes near the residual point (as $k_\mathrm{r}$ rapidly goes to zero). This product can thus be extended at and below the residual point with a value of zero, allowing the computation of the liquid

water flux to be defined at and below the residual point. However, using this technique for the computation of the liquid water





flux presents one main issue. The use of the liquid water content gradient as the driving force for the liquid water flux implies that the state of equilibrium is characterized by a uniform liquid water content rather than uniform water (matric + gravitational) potential. While this is not a problem in the case of a homogeneous medium, it will negatively impact the representation of capillary barriers in a stratified medium, unless a specific treatment is implemented at the interfaces between the different strata
(Amaziane et al., 2012). This is a major issue for stratified snowpack, as capillary barriers are crucial features that need to be captured (Wever et al., 2016). The second technique proposed in the literature to circumvent the singularity of the WRC near the residual point is to regularize it, so that it does not diverge (Beaude et al., 2019; Moure et al., 2023). The matric potential simply assumes a finite value near and below the residual point.

For our article, we rely on the second technique, namely the regularization of the WRC. Concretely, this is done by limiting the divergence of the retention curve up to a critical liquid water content $\theta_{\text{lim}}$ and thus a corresponding limit matric potential $\psi_{\text{lim}}$. Below that point, the liquid water content is allowed to further decrease (down to $0$ in the case of fully dry and cold snow) while the matric potential remains at the value $\psi_{\text{lim}}$. This was done by choosing a unique limit saturation degree $S_{\text{lim}} = \frac{\theta_{\text{lim}} - \theta_{\text{r}}}{\theta_{\text{s}} - \theta_{\text{r}}}$, applied to all snow, below which the WRC reaches its plateau. Some regularized WRC are depicted in Fig. 1 for three different
snow samples. The value of $S_{\text{lim}}$ needs to be taken small (typically $10^{-10}$), in order not to restrict too much the sharp increase of the matric potential at low LWCs. With this modification of the WRC, $\theta_{\text{lim}}$ assumes a role similar to that of the residual value: it corresponds to the point that cannot be further dried with water flow alone. To be consistent with this idea, we propose to also modify the expression of the relative hydraulic conductivity such that it vanishes at $\theta_{\text{lim}}$ rather than $\theta_{\text{r}}$. Specifically, we take

$$
\quad k_{\text{r}} = \begin{cases} \sqrt{S} \left( \frac{\int_{S_{\text{lim}}}^{S} \frac{dS}{\psi}}{\int_{S_{\text{lim}}}^{S_{\text{s}}} \frac{dS}{\psi}} \right)^2 = \sqrt{S} \left( 1 - \left( \frac{1 - S^{\frac{n}{n-1}}}{1 - S_{\text{lim}}^{\frac{n}{n-1}}} \right)^{\frac{n-1}{n}} \right)^2 & \text{if } S > S_{\text{lim}} \\ 0 & \text{otherwise} \end{cases} \tag{10}
$$

which vanishes when $\theta_{\text{w}} = \theta_{\text{lim}}$. As $\theta_{\text{lim}}$ corresponds to the point where liquid cannot flow anymore, we refer to as it as the retention point, to distinguish it from the residual point $\theta_{\text{r}}$. Note that while we have constructed the regularization as a plateau, other choices could be made. For instance, the WRC could be regularized through a linear function (Moure et al., 2023). With the help of regularized WRCs, Richard's equation now applies to describe both dry and wet snow, and can naturally handle
regime changes between them.

### 2.3   Ice budget

Besides the energy and liquid water budgets, the process of water melting and freezing needs to be accounted for in the ice budget as well, and this in order to properly close the mass budget of the snowpack. There are two potential ways to account for melting refreezing event in the ice budget while respecting mass balance: it can be either viewed as a decrease or increase
in density at constant volume, or as a loss or gain of volume at constant density. To the best of our knowledge, the choice in snowpack models is usually to treat refreezing as a density increase and the melting as a thinning of the snowpack, with no





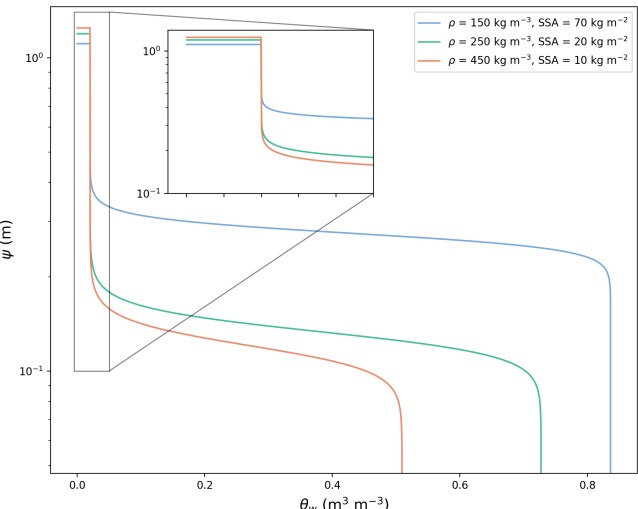

**Figure 1.** Examples of the regularized water retention curves used in this work, for three different snow density and surface specific area (SSA).

direct impact on the density (Bartelt and Lehning, 2002; Vionnet et al., 2012). The justification behind this choice follows the observation that melting snow is usually of a high density, and thus that the melting process should not act as a de-densification mechanism. Yet, as phase changes occur directly within the snow microstructure, at the surface of the porous ice matrix, we rather believe that both melting and refreezing rather impact the snow density, without direct impact on the thickness. With this idea, the relatively high-density nature of melting snow could be attributed to the low viscosity of wet snow (Vionnet et al., 2012) that easily compacts under mechanical stress. Therefore, for the rest of the paper we assume that both type of phase changes affect the snow density, and not directly the snow layer thicknesses.

The ice mass budget within the snowpack is thus given by

$$\rho_i \partial_t \phi_i = M_{\text{freeze}} \tag{11}$$

where $\phi_i$ is the ice volume fraction and $\rho_i$ the ice density (in $\text{kg m}^{-3}$).

## 2.4 Resulting equations

Replacing Eq 5 by its sum with $\rho_w L_{\text{fus}}$ Eq 6, we can rewrite the system of equation as:



$$
\quad
\begin{cases}
\partial_t h - \nabla \cdot (\lambda \nabla T + \rho_\mathrm{w} L_\mathrm{fus} \nabla \cdot (K^\mathrm{sat} k_\mathrm{r} (\nabla \psi + \cos(\gamma)))) - Q_\mathrm{abs} = 0 \\
\rho_\mathrm{w} \partial_t \theta_\mathrm{w} - \rho_\mathrm{w} \nabla \cdot (K^\mathrm{sat} k_\mathrm{r} (\nabla \psi + \cos(\gamma))) + M_\mathrm{freeze} = 0 \\
\partial_t m_\mathrm{i} - M_\mathrm{freeze} = 0
\end{cases}
\tag{12}
$$

where $h = h_\mathrm{s} + \rho_\mathrm{w} L_\mathrm{fus} \theta_\mathrm{w} = c_\mathrm{p} (T - T_0) + \rho_\mathrm{w} L_\mathrm{fus} \theta_\mathrm{w}$ is the sum of energy carried by the snow and the liquid water within. The first equation corresponds to the total energy budget. It includes the processes of heat conduction and advection of energy with liquid water as fluxes, as well as the volumetric source of energy. The release/absorption of latent heat is not included as a source term in this equation. Indeed, such phase change does not change the total energy budget, as latent heat is converted into snow sensible heat, both included in $h$.

In its current form, this system of equation includes 6 unknown, namely $h$, $T$, $\psi$, $\theta_\mathrm{w}$, $M_\mathrm{freeze}$ and $\phi_\mathrm{i}$ ($K^\mathrm{sat}$, $k_\mathrm{r}$, and $\lambda$ being material properties that can be deduced from these unknowns). To this system of equation can be added 2 constitutive laws: Eq. 3 relating the snow temperature to its sensible heat and Eq. 8 relating the matric potential to the liquid water content.

The system of equations is thus not yet closed, as it remains to express how $M_\mathrm{freeze}$ relates to the other variables. To the best of our knowledge, there is no broadly accepted theoretical or experimental work providing the rate of melting or freezing of water in snow (Moure et al., 2023). As most snowpack models (Jordan, 1991; Bartelt and Lehning, 2002; Vionnet et al., 2012; Sauter et al., 2020), we assume that liquid water and the snow are in local thermodynamic equilibrium. Specifically, it is assumed that snow below the fusion temperature and liquid water cannot coexist. This assumption is consistent with the slow matric flow considered in this article. In this framework, $h$ and $\theta_\mathrm{w}$ become directly related: in the case where $h$ is below the fusion value, $\theta_\mathrm{w}$ vanishes; in the case where $h$ is above the fusion value, $\theta_\mathrm{w}$ is proportional to the excess of energy. In this case, the liquid water content and the temperature can be expressed as a function of the total energy:

$$
\theta_\mathrm{w} =
\begin{cases}
0 & \text{if } h < 0 \\
\frac{h}{\rho_\mathrm{w} L_\mathrm{fus}} & \text{otherwise}
\end{cases}
\tag{13}
$$

and

$$
\quad T =
\begin{cases}
T_0 + \frac{h}{c_\mathrm{p}} & \text{if } h < 0 \\
T_0 & \text{otherwise}
\end{cases}
\tag{14}
$$

The rate of freezing/melting can then be derived, as a diagnostic, from the closure of Richard's equation. It physically corresponds to the amount of frozen and melted water required to maintain the local thermodynamic equilibrium. It can also be eliminated from the system of equations by summing Eqs 6 and 11, simply resulting in:





$$\begin{cases} \partial_t h - \nabla \cdot (\lambda \nabla T + \rho_{\mathrm{w}} L_{\mathrm{fus}} \nabla \cdot (K^{\mathrm{sat}} k_{\mathrm{r}} (\nabla \psi + \cos(\gamma)))) - Q_{\mathrm{abs}} = 0 \\ \rho_{\mathrm{w}} \partial_t \theta_{\mathrm{w}} + \rho_{\mathrm{i}} \partial_t \phi_{\mathrm{i}} - \rho_{\mathrm{w}} \nabla \cdot (K^{\mathrm{sat}} k_{\mathrm{r}} (\nabla \psi + \cos(\gamma))) = 0 \end{cases}$$
(15)

with the second equation now corresponding to the total mass budget.

Finally, we note that the recent article of Moure et al. (2023) proposes to relax the assumption of local thermal equilibrium and to introduce a finite rate of phase change, derived from the upscaling of the Frenkel-Wilson equation. This implies that
the ice and liquid water temperatures in a snow sample are in general different and can be above or below the fusion point. This modeling framework, composed of four partial differential equations, is briefly presented in Appendix A. However, as observed in the Appendix, the timescale of relaxation towards local thermodynamical equilibrium appears to be much smaller than the timescale of matric water movement and heat diffusion, supporting the standard assumption of local equilibrium in snowpack models.

**2.5 Choosing primary variables in the presence of regime changes**

A difficulty with the system presented in Eq 15 (complemented with its material laws) is the presence of 4 distinct regimes, where parts of the equations behaves differently and sometimes degenerate. The first regime corresponds to dry snow, when the temperature is below the freezing point. Here, the thermodynamical state of snow can be characterized by either its temperature or its energy content (the two being related by the thermal capacity). To the contrary, the LWC and the matric potential cannot
be used to describe the state of snow, as they assume constant, degenerate, values. In the second regime, the snow is above its fusion point, but the LWC remains below its retention value. Here, the snow can be characterized by its energy content or its LWC, but not by its temperature nor its matric potential. The third regime corresponds to the classical unsaturated Richards' case, where the LWC is above the retention point. Here, the snow can be characterized by its energy content, its LWC, or its matric potential, but not by the temperature. The fourth regime corresponds to the water-saturated regime. Here, the snow
can only be characterized by its matric potential. This regime corresponds to the saturated regime of Richards' equation. A summary of the different regimes and the variables that can be used to characterize them is given in the Table 1. As seen, there is no natural variable that can be used to describe the thermodynamical state of snow over the 4 different regimes. While the energy content could be used to characterize snow in regimes 1, 2, and 3, it degenerates in regime 4. Similarly, while the matric potential is a good candidate to describe regimes 3 and 4 (as usually done when numerically solving Richards' equation; Celia
et al., 1990; Farthing and Ogden, 2017), it fails in regime 1 and 2.

To circumvent this issue, we rely on a variable switch technique by parametrization (Brenner and Cancès, 2017), through the use of a fictitious variable meant to behave as the energy content in the first three regimes and as the water matric potential



**Table 1.** Summary of the various regimes encountered in snow. For each regime, the relevant variables that can be used to characterize snow are given, as well as the degenerated variables that cannot be used to characterize snow. In all cases, the ice volume fraction $\phi_i$ is also needed as a second variable.

|  | Dry snow | Wet non-flowing snow | Wet unsaturated snow | Wet saturated snow |
|---|---|---|---|---|
| Relevant variables | h, T | h, $\theta_w$ | h, $\theta_w$, $\psi$ | $\psi$ |
| Degenerated variables | $\theta_w$, $\psi$ | T, $\psi$ | T | T, h, $\theta_w$ |

in the last one (Bassetto et al., 2021, 2022). For this, we introduce a new variable $\chi$ (expressed in $\mathrm{J\,m^{-3}}$) used to parameterize the $\{h, \psi\}$ graph. Specifically, we choose $\chi$ such that:

$$
h(\chi, \phi_i) = \begin{cases} \chi & \text{if } \chi < \theta_s(\phi_i)\rho_w L_{fus} \\ \theta_s(\phi_i)\rho_w L_{fus} & \text{otherwise} \end{cases} \tag{16}
$$

$$
\psi(\chi, \phi_i) = \begin{cases} \psi(\theta_w(\chi, \phi_i)) & \text{if } \chi < \theta_s(\phi_i)\rho_w L_{fus} \\ \frac{\chi - \theta_s(\phi_i)\rho_w L_{fus}}{\beta} & \text{otherwise} \end{cases} \tag{17}
$$

where $\beta$ is an arbitrary constant, introduced to respect the unit homogeneity of Eq 17, and where the dependence of $\phi_i$ as also

been made explicit. Thanks to this variable $\chi$, we are now able to characterize the energy content and matric potential of snow in all regimes, and from it to derive the values of all other relevant quantities for snow ($\theta_w$, $T$, $k_r$, etc).

Once parameterized with $\chi$ our system of equation becomes:

$$
\begin{cases} \partial_t h(\chi, \phi_i) - \nabla \cdot (\lambda \nabla T(\chi, \phi_i) + \rho_w L_{fus} \nabla \cdot (K^{sat} k_r(\nabla \psi(\chi, \phi_i) + \cos(\gamma))) - Q_{abs} = 0 \\ \rho_w \partial_t \theta_w(\chi, \phi_i) + \rho_i \partial_t \phi_i - \rho_w \nabla \cdot (K^{sat} k_r(\nabla \psi(\chi, \phi_i) + \cos(\gamma))) = 0 \end{cases} \tag{18}
$$

that can be solved searching for $\chi$ and $\phi_i$. We have explicitly shown the dependency of $h$, $T$, $\psi$, and $\theta_w$ on both $\chi$ and $\phi_i$, but one should keep in mind that the effective properties $\lambda$, $K^{sat}$, and $k_r$ also depends on the value of $\chi$ and $\phi_i$. With this set of equations, we now have a consistent description of the energy and mass budgets in snowpacks, that naturally applies to both dry and wet snow.

## 3 Numerical implementations

The goal of this section is to present and explore different numerical implementations of the energy and mass budgets in snow, in order to investigate the impacts of the implementation on the results and robustness of the models. We decided to focus this





comparison on (i) the use of a regularized WRC with variable switching for matric flow, and (ii) the loosely or tightly-coupled nature of the implementation, where processes can either be solved sequentially (i.e. with operator-splitting) or concomitantly (i.e. with tight-coupling) (Steefel and McQuarrie, 1996; Keyes et al., 2013) . By varying the regularization of the WRC and

the degree of operator-splitting, we have implemented five snowpack toy-models. Below, we start by briefly presenting the common features shared by all implementations, and then discuss the concrete differences between them. These differences between the numerical implementations are summarized in Table 2.

## 3.1 A common core between all models

In order to be directly comparable, the models share portions of their numerical implementations. This includes compaction under mechanical stress (assuming a linear viscosity between stress and deformation and which is solved after the thermodynamics), the constitutive laws defining the material properties of snow, the spatial and temporal discretization, and the method for solving the resulting non-linear systems of equations.

### 3.1.1 Constitutive laws

To be fully closed, the equations of energy and mass conservation need to be complemented with constitutive laws, prescribing the material properties of the snow material. We assume the thermal conductivity $\lambda$ of snow to follow Calonne et al. (2011) and the saturated hydraulic conductivity $K^{\mathrm{sat}}$ to follow Calonne et al. (2012). The WRC (Eq. 8) and the relative hydraulic conductivity (Eq. 10) require defining the parameter $\alpha$ and $n$. For this, we follow Yamaguchi et al. (2012). The residual LWC $\theta_{\mathrm{r}}$ is taken as $0.02$ (Yamaguchi et al., 2010; Wever et al., 2014; D'Amboise et al., 2017) and the saturated LWC $\theta_{\mathrm{s}}$ as $1 - \phi_{\mathrm{i}}$

(i.e. the total porosity). Finally, for the compaction of the snowpack under mechanical settling, we use the linear compactive viscosity of Vionnet et al. (2012), including its dependence to the LWC. The precise expressions for the constitutive laws are given in Appendix B.

### 3.1.2 Finite Volume discretization

To be numerically solved, the system of Eqs. 18 needs to be discretized in time and space. For the time discretization, we use a

simple backward Euler time-stepping. This choice is motivated by the overall stability of the method, mitigating the apparition of overshoots and oscillations in the case of large-time steps (Butcher, 2008). For the spatial discretization, we use a finite volume scheme. Briefly, this numerical method relies on dividing the snowpack into $N$ cells and performing energy and mass balances on each individual cell. In this framework, each cell is described by its average $\chi$ and $\phi_{\mathrm{i}}$ values. The evolution of these average values is obtained by integrating the system of equations over the different cells, and applying the divergence theorem

to transform the divergence operator into fluxes at the cells' boundaries.





To be mathematically closed, the heat and liquid water fluxes at the interface between two adjacent cells need to be reconstructed from the cells' averages. For the heat fluxes, this is done by computing the temperature gradient from one cell center to the other, and defining an interfacial thermal conductivity. As classically done in finite volume schemes, this interfacial thermal conductivity $\lambda^{k+1/2}$ (between cells $k$ and $k+1$) is taken as some average of the thermal conductivities of the adjacent cells. In our case, we take it as the weighted harmonic average. This choice ensures that the heat flux vanishes when the thermal conductivity of one of the two cells vanishes (Kadioglu et al., 2008).

For the computation of the liquid water, we discretize as well the matric potential gradient using the average values in the cell and the center-to-center distance. For the computation of the interfacial hydraulic conductivity, we use the so-called upstream mobility formulation (Bassetto et al., 2021). In this framework, the hydraulic conductivity of the interface is split into the product of the saturated conductivity $K_{\mathrm{sat}}^{k+1/2}$ and the relative conductivity $k_{\mathrm{r}}^{k+1/2}$. As with the thermal conductivity, the interfacial saturated conductivity is taken as the harmonic average of the saturated conductivities of the neighboring cells. This ensures that the liquid water flux vanishes when on the of the cell is impermeable. The interfacial relative conductivity $k_{\mathrm{r}}^{k+1/2}$ between cell $k$ and $k+1$ is taken as the upstream value, i.e.:

$$
k_{\mathrm{r}}^{k+1/2} = \begin{cases} k_{\mathrm{r}}^{k} & \text{if } \psi^{k} > \psi^{k+1} + \cos(\gamma)d^{k+1/2} \\ k_{\mathrm{r}}^{k+1} & \text{otherwise} \end{cases} \tag{19}
$$

where $d^{k+1/2}$ is the distance between the neighboring cell centers and $\gamma d^{k+1/2}$ thus is the vertical distance between the cell centers. With this upstream choice, the numerical scheme becomes monotonic (in the sense of Eq. (3.18) of Bassetto et al., 2021). This property is notably beneficial for the convergence of the Newton (or other iterative) method, when solving the resulting non-linear system of equations.

### 3.1.3 Truncated Newton method with adaptive time-stepping

The systems of equations to be numerically solved in the different models are non-linear and require an iterative scheme. For that, we rely on a Newton method, with a stopping criterion when the iterated estimations is close enough to the solution. In models 4 and 5, this criterion is complemented with a criterion on mass conservation, as these numerical scheme are not naturally mass-conservative. While the Newton algorithm provides a relatively fast convergence when the starting estimation is close to the solution, it is not a globally convergent algorithm. In other words, it is possible that the algorithm diverges or get stuck in cycles. To improve its convergence performance, we implemented two strategies. First, as usually done with Richards' equation, we use an adaptive time-step. In the case where the algorithm fails to converge after a given number of iterations, the algorithm is rewind to the start of the time-step and its value halved. By default, we set the maximum number of iterations to 25. In the rest of the article, we thus make the difference between the so-called "default timestep", corresponding to the timestep that is ideally used if no convergence problems are encountered, and the "adaptive timestep", that is effectively used to solve the nonlinear equations involving matric flow. Secondly, we use the so-called truncation method when a transition from one regime to another occurs (Bassetto et al., 2020). Indeed, the transitions between regimes are corner points, characterized





by discontinuities in the derivatives. Using the derivative computed on one side of a corner point to derive the evolution of the estimate in the other side is therefore problematic and can lead to overshoots, impeding the convergence towards the solution. To avoid this problem, each time a transition from one regime to another occurs in a cell of the snowpack, $\chi$ is set back in the vicinity of the corner point. In practice, $\chi$ is placed at a small $\epsilon$ value before or after the corner point, in order to fall in the good regime. We set this value of $\epsilon$ to be $10^{-5}\,\mathrm{J\,m^{-3}}$. Finally, we note that for solving Richards' equation the SNOWPACK (Wever et al., 2014) and Crocus (D'Amboise et al., 2017) models decided to follow a modified Picard method (Celia et al., 1990), where the contribution of the change in hydraulic conductivity in the Jacobian is not taken into account. As a test, we also run simulations using the modified Picard rather than the Newton method for models 4 and 5. This did not modify the results discussed later.

### 3.2 Differences between the five different numerical implementations

### 3.3 Model 1: Fully coupled with a regularized WRC

For the first model, we use the regularized WRC and solve the energy and mass budgets in a tightly-coupled manner, that is to say that the system of Eqs 18 is solved concomitantly for both $\chi$ and $\phi_\mathrm{i}$. In this version, there is therefore no degree of operator-splitting when solving the thermodynamical state of the snowpack.

### 3.4 Model 2: Partially coupled with a regularize WRC

While the numerical implementation of model 1 appears to be the most consistent, it results in a large non-linear system that can be numerically expensive to solve. As a compromise, some degree of operator-splitting (i.e. sequentiallity) between processes can be introduced. For model 2, we thus used the regularized WRC with a decoupling of the computation of the energy and liquid water budgets from that of the ice budget. The motivation behind it is that the timescale for the evolution of the density is usually longer (of the order of the day) than that of the energy and liquid water evolution (of the order of the hour or less). Concretely, in Eq.18 we thus first solve the evolution of $\chi$ (assuming a fixed $\phi_\mathrm{i}$). Then, we find the evolution of $\phi_\mathrm{i}$ by closing the mass budget.

### 3.5 Model 3: Full operator-splitting with a regularized WRC

Keeping with the idea of using operator-splitting to reduce the numerical cost and complexity of the model, the third implementation is based on a decoupling of the energy, liquid water, and ice budgets solving. Specifically, we first evolve $\chi$ under the process of heat conduction and phase changes only. We then, evolve $\chi$ again, this time under the process of matric liquid water flow. Finally, $\phi_\mathrm{i}$ is updated by re-applying phase transition and closing the mass budget.

### 3.6 Model 4 and 5: Full operator-splitting without a regularized WRC

The last two models are meant to emulate the techniques used so far in snow models to handle the degeneracy of the retention curve in dry snow. Specifically, model 4 is based on the SNOWPACK implementation (Wever et al., 2014) and model 5 on



**Table 2.** Description of different implemented toy-models based on (i) the regularization of the WRC, (ii) whether the primary variable is a switch (behaving as $h$ in unsaturated snow and $\psi$ in saturated snow) or $\psi$ for solving the matric flow process, and (iii) the degree of couplings between thermodynamics processes.

|  | Model 1 | Model 2 | Model 3 | Model 4 and 5 |
|---|---|---|---|---|
| WRC | Regul. | Regul. | Regul. | Non Regul. |
| Primary variable for liquid flow | Switch | Switch | Switch | $\psi$ |
| Degree of coupling | - Conduction, phase change, and liquid flow coupled. | - Conduction and phase change coupled. <br> - Liquid flow decoupled | - Conduction, phase change, and liquid flow decoupled | - Conduction, phase change, and liquid flow decoupled |

**Table 3.** Description of three different test cases considered in this article, with their durations and external forcings.

|  | Test case 1 <br> Melting snowpack | Test case 2 <br> Low-intensity rain | Test case 3 <br> High-intensity rain |
|---|---|---|---|
| Duration | 6 days | 1 day | 2 day |
| Shortwave | $840\,\mathrm{W\,m^{-2}}$ peak | $420\,\mathrm{W\,m^{-2}}$ peak | $420\,\mathrm{W\,m^{-2}}$ peak |
| Longwave and Turbulent fluxes | $200$ to $310\,\mathrm{W\,m^{-2}}$ | $310\,\mathrm{W\,m^{-2}}$ | $310\,\mathrm{W\,m^{-2}}$ |
| Rainfall rate | No rain | constant $1\,\mathrm{mm\,h^{-1}}$ | $15\,\mathrm{mm\,h^{-1}}$ for $4\,\mathrm{h}$ |

the Crocus implementation (D'Amboise et al., 2017). The specificities of the implementation are given in Appendix C, but

we recall that both techniques are based on the idea of introducing a small amount of liquid water in cold snow, shifting the residual liquid water content value below it, and using $\psi$ as the primary variable. Moreover, these models rely on a decoupled resolution of the thermodynamical processes.

# 4   Numerical tests

The goal of this section is to build simple synthetic examples, meant to represent different situations in which liquid water percolation might occur in snowpacks, and to investigate the differences in behavior between the various implementations.





### 4.1 Test case 1: Melting snowpack

The first test case is meant to represent the situation of melting snowpack, releasing liquid water near the surface that then percolates downward. For the initialization of the snowpack, we rely on a simulation performed with the Crocus snowpack (Vionnet et al., 2012). This yields a realistic stratigraphy, with thinner cells near the surface in order to better capture surface effects. The initial state of the simulation was chosen to obtain a dry snowpack near its peak snow water equivalent. This defines the ice volume fraction, temperature, and specific surface area at the start of our simulations. Note that the specific surface area

is needed as it plays a role in the hydraulic properties of the snowpack (Calonne et al., 2012; Yamaguchi et al., 2012). However, as we did not implement metamorphism laws in our toy model, the specific surface area is simply kept constant during the simulation.

For the top boundary conditions, we impose an incoming diurnal energy flux at the surface of the snowpack (oscillating between $310\,\mathrm{W\,m^{-2}}$ at noon and $200\,\mathrm{W\,m^{-2}}$ at night), encapsulating the effect of incoming longwave radiations and turbulent

heat fluxes. We also impose diurnal shortwave radiations (peaking at $840\,\mathrm{W\,m^{-2}}$ at noon and vanishing at night), that penetrate within the snowpack (with an albedo of $0.7$ and an e-folding depth of $5\,\mathrm{cm}$; Libois et al., 2013). No liquid, nor solid, precipitation is considered in this test case. As the bottom boundary condition, we impose a constant heat flux of $10\,\mathrm{W\,m^{-2}}$, emulating the heat flux received from a warm ground below, and a free-drainage condition for the liquid water flux. For each model, simulations are run for 6 days, so that water percolates to the bottom of snowpack. Default timesteps vary between 112

and $7200\,\mathrm{s}$ (we recall that because of the adaptive timestep strategy, the actual timestep used for some process might be shorter than the prescribed default value).

### 4.2 Test case 2: Water infiltration under low-intensity rain

This second test case is meant to represent the slow infiltration of rain under a long but low-intensity event. The initialization is the same as in case 1, based on the output of the same Crocus simulation. For the top boundary condition, we impose a

constant surface energy flux of $310\,\mathrm{W\,m^{-2}}$, a shortwave radiation flux peaking at $420\,\mathrm{W\,m^{-2}}$ at noon (lower than in the first test case to represent cloudiness), and a constant rain flux of $1\,\mathrm{mm\,h^{-1}}$ lasting the whole simulation. For the bottom boundary condition, we use the same conditions as in the test case 1 (constant heat flux and free-drainage). Simulations were performed with each model for 1 day, in order for shallow infiltration to occur. Default timesteps range from 112 to $7200\,\mathrm{s}$.

### 4.3 Test case 3: Water infiltration under high-intensity rain

The third test case represents the case of a short but high-intensity rain event. The initialization and boundary conditions are taken as in the test cases 2, except for the incoming rain flux that is now of $15\,\mathrm{mm\,h^{-1}}$ and lasts for $4\,\mathrm{h}$. This results in an abrupt input of liquid water in the snowpack, rapidly percolating downward. Simulations were run for about 2 days, long enough for deep percolation to occur. Default timesteps range from 112 to $7200\,\mathrm{s}$.



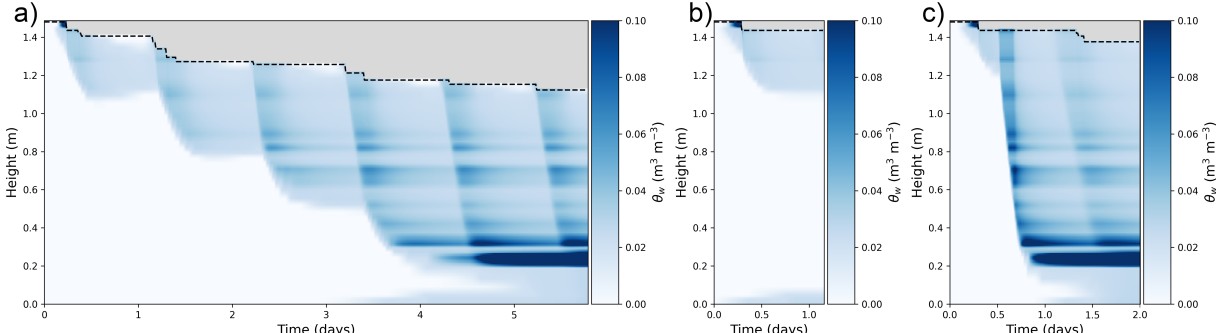

**Figure 2.** Results of the reference simulations for the three investigated test cases: a) surface melting with a deep infiltration of surface melt, b) a low-intensity rain with limited infiltration, c) a high-intensity rain event with a fast and deep water infiltration.

For all three test cases, reference simulations were performed with the fully coupled model (model 1) and a timestep of $1\,\text{s}$. The resulting LWC profiles over time are displayed in Fig. 2.

# 5 Results and discussion

## 5.1 Robustness and numerical cost

The first difference in behavior between the different implementations is their respective numerical cost, i.e. the computation time required to perform a given simulation. In the presence of an adaptive timestep, required to accommodate the strong non-linearity of the liquid matric flow and of the WRC, we observed that the numerical cost of the models is, at first order, controlled by the ability of the models to stick to large adaptive timesteps. Figure 3 displays the adaptive timesteps required by the models in the three test cases presented above. While the models using a regularized retention curve with a variable switch (models 1, 2, and 3) are able to maintain a timestep of around $1\,\text{h}$ for most of the simulations, models 4 and 5 requires the adaptive timestep to regularly drop, sometimes below $10\,\text{s}$. In the end, models 4 and 5 require about one order of magnitude more steps to perform the same simulation, resulting in significantly increases numerical cost and computation time. For instance, in the case of the test case 3 with a default timestep of $3600\,\text{s}$, models 4 and 5 require a computation time of about $500\,\text{s}$, while models 1, 2, and 3 require about $130\,\text{s}$, $25\,\text{s}$, and $45\,\text{s}$, respectively. Note that these numbers should only be interpreted qualitatively to assess numerical complexity, as the precise computation time is also influenced by factors that we did not control for (such as the level of code optimization or memory-caching). Therefore, the use of a regularized WRC with a variable switch appears as a very favorable practice to increase the robustness of the numerical scheme and to decrease its numerical cost by accommodating larger timestesp. Our understanding for this behavior is that the combination of regularized WRC and not using the matric potential as the primary variable near the retention point hinders the presence of abrupt, and nonlinear, variations of the numerical solution, thus making the problem easier to solve. To the contrary, using the matric potential as the primary variable is known to be inefficient for low LWC materials and to require timesteps that are much smaller than the



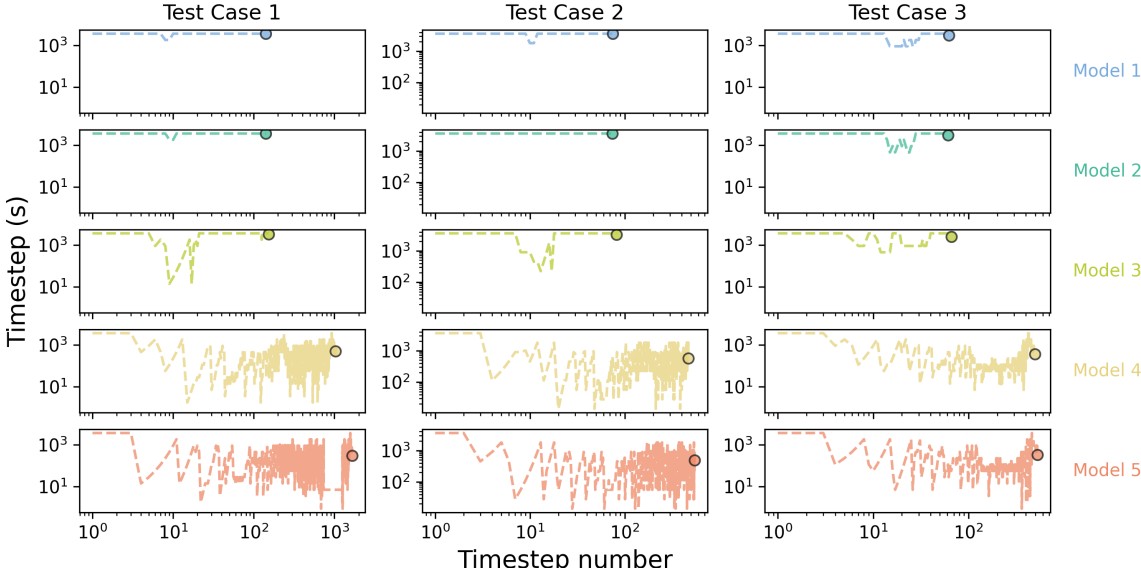

**Figure 3.** Adaptive timesteps required to simulate matric flow in each test case (columns) and for each model implementation (rows). The default value for the timestep is set to 3600 s. In each plot, the marker marks the total number of timesteps required and the average timestep size.

actual timescale of liquid water transport (Forsyth et al., 1995; Sadegh Zadeh, 2011).

Focusing on models 1, 2, and 3 in Fig. 3, it appears that these three implementations require very similar timesteps, apart from a notable drop for model 3 in the first and second test case. Their differences in numerical cost should be controlled by the complexity of the assembly and solving of the matrix-form of the equations. According to our numerical test cases, model 2 and 3 have about the same numerical cost, while our implementation of model 1 requires a longer computational time. This is to be expected, as model 1 requires the largest matrix to be assembled and solved ($2N \times 2N$), while the other model handle smaller matrices ($N \times N$). However, we also note that a large portion of the up-cost of model 1 comes from the computation of derivatives with respect to $\phi_i$ for the assembly of the Jacobian. In our implementation, computing these derivatives comes with a high cost, as we rely on generic automatic differentiation without any attempt at optimization. If these derivatives with respect to $\phi_i$ are ignored (thus transforming the scheme from a true Newton to a form of modified Picard method), the numerical solving of the coupled system of Eqs.18 can be efficiently cheapen. For instance, in the aforementioned test case 3 with a default timestep of 3600 s, the computational time can be lowered from 130 to about 50 s, i.e. in the same range as models 2 or 3.





## 5.2 Error-level and timestep convergence

Besides their numerical cost, the different implementations also yield different levels of error when compared to the reference simulations. While the error level generally tends to decrease with shorter timesteps, not all models perform the same for decreasing timesteps. Here, we thus investigate the degree error of each model, for default timesteps ranging from 7200 to 112 s. For that, we compute the Root Mean Square Difference (RMSD) between the models' outputs and the reference simulations. Figure 4 display the RMSDs in LWC and temperature for the three test cases as a function of the default timestep. Note that the RMSD is computed over the whole snowpack, while errors tend to be spatially located near the percolation front. Therefore, the RMSD value is smaller, typically of an order of magnitude, than the errors occurring at the percolation front. As expected, the overall trend is a decrease of the RMSDs with small timestep. However, while the unregularized models 4 and 5 perform similarly as the other models with large default timesteps, they do not show a clear decrease of error at smaller timesteps. Our understanding is that the strategy of modifying the residual LWC based at the start of each timestep impacts the physics and changes the solution to which the models converge with small timesteps. This results in a plateau or even an increase of the associated RMSDs in Fig. 4. Concerning the regularized models (1, 2, and 3), our results suggest that model 1 consistently yields the lowest RMSDs, for timesteps of 900 s and less. The comparison of the errors of models 2 and 3 depends on the specific test case and the variable of interest, with no clear advantage of one over the other. We note that under some circumstances (e.g. test case 2 and timestep of 900 s), one model can appear better in a metric (e.g. LWC RMSD) while it is outperformed in another metric (e.g. temperature RMSD).

## 5.3 Is full-coupling benificial ?

As seen above, the use of a tightly-coupled resolution in numerical models usually provides a slightly better precision and a better robustness (Keyes et al., 2013; Brondex et al., 2023), at the expense of a more complex numerical resolution. Indeed, by ensuring physical consistency between the different variables, tightly-coupled resolutions are known to prevent overshooting, which could develop into numerical oscillations or even divergence (Brondex et al., 2023). One of the motivation behind the investigation of the impact of operator-splitting on the numerical resolution of the energy and mass budgets with liquid water matric flow was thus to examine whether a tightly-coupled resolution was warranted in this case. To our surprise, the problem of heat conduction, liquid matric flow, and phase changes, appears to be quite robust to operator splitting. Indeed, even in the case of a quite large timestep of 4 hr, we did not observe any clear sign of numerical instabilities or very large overshoots in the models using operator-splitting. Our understanding is that this overall insensibility to operator-splitting comes from the nature of the involved processes, which do not play in the same regimes. Indeed, the process of heat conduction only occurs in dry snow, the process of matric flow in wet snow, and the process of phase change at the wet-dry interface. The possibility of interaction between processes thus remains limited and the large physical inconsistencies, that can be problematic with operator-splitting, did not emerge in most situations. Therefore, the standard practice of operator-splitting seems to be well-justified, at least when representing the interaction of matric flow with heat conduction and phase changes in snowpack models.




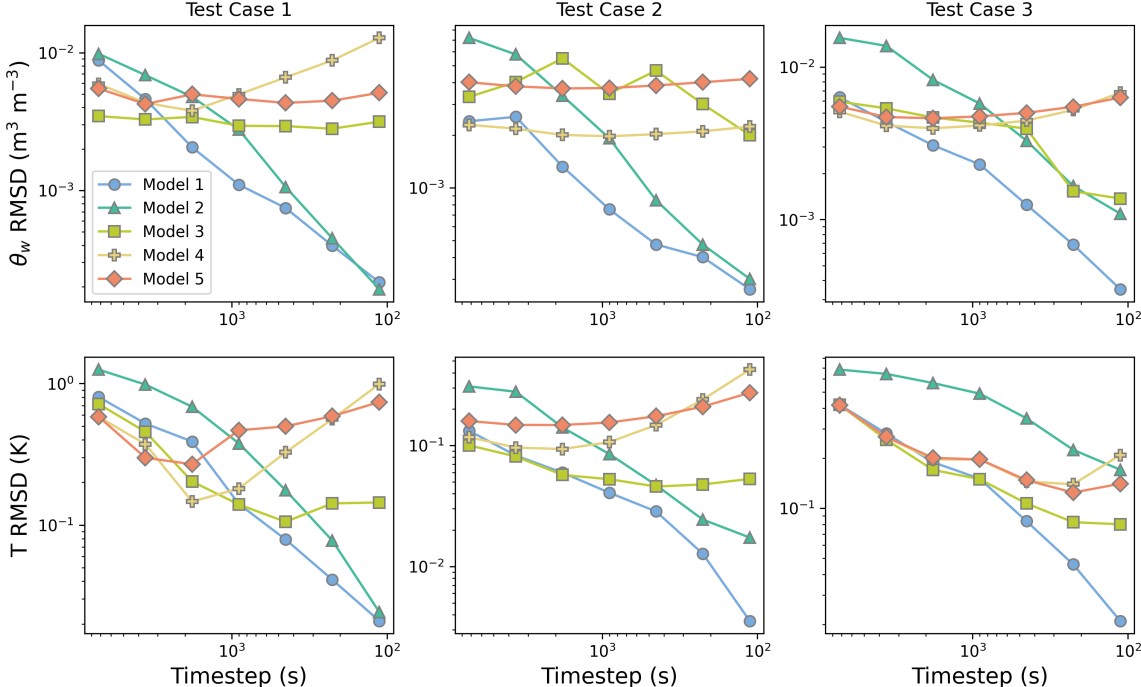

**Figure 4.** Root Mean Squared Difference in LWC and temperature between the models and a reference simulation as a function of the default timestep size.

## 5.4 Inclusion of other physical mechanisms

In this article, we considered the coupled mechanisms of heat conduction, matric liquid water percolation, and liquid-solid phase changes. While we focused on these mechanisms, as they can be found in all snowpack models (with more or less degrees of complexity), others thermodynamical mechanisms could be included as well. This is for instance the case of water vapor diffusion and vapor phase changes. To the best of our understanding, vapor-related mechanisms could be easily added to the systems of the thermodynamical equations governing snowpacks. For that, one would need to (i) introduce a new partial differential equation governing the mass balance of water vapor, (ii) introduce a water vapor component in the energy balance equation, and (iii) close the system by prescribing the phase change term for the gas phase (or assuming water vapor thermodynamical equilibrium with the other phases) and parametrizing the diffusivity of vapor. This could be done based on works of Calonne et al. (2014), Jafari et al. (2020), Brondex et al. (2023), and Bouvet et al. (2024).

Similarly, an important mechanism relating to the transport of liquid water in snowpacks is the process of preferential (i.e. non-matric) flow (Schneebeli, 1995; Yamaguchi et al., 2018). While the exact mechanisms, and therefore description, of preferential flow in snowpacks remain unclear at this point, the presence of fast flowing, and out-of-equilibrium with the rest of the snow layer, liquid water appears as a prerequisite for the formation of internal ice layers (Fierz et al., 2009). As percolation chimneys are likely to play a key role for preferential flow, the use of a so-called dual-porosity/dual-permeability model (e.g.



Vogel et al., 2000; Lewandowska et al., 2004) has notably been proposed to take into account preferential flow in the SNOW-PACK model (Wever et al., 2016; Quéno et al., 2020). As preferential flow share many similarities with matric flow (being essentially a faster version), the governing equations are subjected to the same degenerate behaviors in the case of a dry or

a fully-saturated medium. Therefore, the techniques explored in this article, namely the regularization of the WRC to handle dry media with the use of a switch variable, could also benefit the numerical implementation of preferential flow. Moreover, as preferential flow is a fast and out-of-equilibrium process, it can introduce stiffness in the equations governing snowpacks (Fazio, 2001), potentially resulting in overshoots and oscillations when using operator splitting with a large timestep (Brondex et al., 2023; Fourteau et al., 2024). As discussed above, the derivation of a fully-consistent system of equations, that applies

from dry to water-saturated snow, enables a tightly-coupled numerical resolution, which could be key to handle stiff and fast systems of equations (Keyes et al., 2013).

## 6   Conclusions

This article focuses on the numerical implementation of matric water flow and its interaction with other physical processes

in snowpack models using Richards' equation. While the use of Richards' equation can improve the representation of water flow, its complexity and numerical cost can sometimes hinder widespread adoption. To overcome this issue, we explored a new numerical implementation of Richard's equation. We started by recalling the governing equations of energy and mass conservation in snowpacks, that govern the evolution of the snow material both in dry and wet zones. However, for these equations to be directly applicable for both dry and wet snow, it is necessary to provide an expression of the liquid water flow

that also applies in dry snow (where simply no flow is expected). For this, we regularized the water retention curve, in order to avoid the divergence of the matric potential as the snow material dries. Another issue with the concomitant and consistent representation of a whole snowpack is that dry and wet snow are not described by the same thermodynamical variables, as dry snow is described in terms of temperature/energy while wet snow is rather described in terms of liquid water content/matric potential. To seamlessly handle this change in the physical variables characterizing the snow material, we introduced a variable

switch technique implemented thanks to the introduction of a fictitious variable building on the idea introduced in Brenner and Cancès (2017). This defines a single thermodynamical variable to describe all states of snow, from dry to water-saturated. Eventually, we thus obtained a consistent system of two equations (energy and mass conservation) governing the evolution of snowpacks and which allows a unified treatment of dry and wet snow.

To compare the behavior and performance of this new description, we have implemented five toy snowpack models. This

includes three models, all using a regularized WRC with a variable switch but relying on various degrees of operator splitting during the resolution, as well as two models based on the treatment of Richards' equation in the SNOWPACK and Crocus snow models (i.e. using a non-regularized retention curve). Based on three test cases, representing various situations of snowpack humidification, we observe that the use of a regularized WRC with variable switch significantly increases the robustness of the numerical implementation, which can run with timesteps of 15 min and above, while the non-regularized models require smaller





timesteps to handle liquid water percolation. On the other hand, the possibility of a fully tightly-coupled implementation seems to have a limited impact compared to an operator-splitting implementation, both in terms of stability and timestep sensitivity. While this article focused on matric liquid flow, we believe that the methodology put forward, namely the use of regularized laws, variable switch, and physically consistent description of the various thermodynamical states of snow, could also be useful for the description of preferential flow. Indeed, as preferential flow is usually modeled as a faster version of matric flox, the

ability of regularized models to nonetheless handle relatively large timesteps could prove crucial to limit their numerical cost.

**Appendix A: Ice-Water thermodynamics equilibrium in snow**

In this section, we briefly discuss the relaxation of the thermodynamics equilibrium assumption between the ice and water phases in snow. This appendix is largely based on the recent work of Moure et al. (2023). If we assume that the ice and liquid water are out-of-equilibrium, it is necessary to distinguish between the ice temperature and the liquid water temperature. It is

thus no-longer possible to write a single energy-equation, as in Eq. 15. Similarly, we also need to distinguish between the solid and liquid mass budgets. The out-of-equilibrium ice and liquid water system is now described by $T_i$, $T_w$, $m_i$, and $m_w$ (the ice and liquid temperatures and the ice and liquid mass content, expressed in $\mathrm{kg\,m^{-3}}$) and the four equations provided by Moure et al. (2023):

$$
\begin{cases}
\partial_t(c_i m_i T_i) + \nabla \cdot J_{u,i} = W_{\mathrm{SSA}} \dfrac{\lambda_i}{r_i}\left(T_{\mathrm{int}} - T_i\right) \\[2mm]
\partial_t(c_w m_w T_w) + \nabla \cdot (J_w c_w T_w) + \nabla \cdot J_{u,w} = W_{\mathrm{SSA}} \dfrac{\lambda_w}{r_w}\left(T_{\mathrm{int}} - T_w\right) \\[2mm]
\partial_t m_i = -\rho_i R_m W_{\mathrm{SSA}}\left(T_{\mathrm{int}} - T_0\right) \\[2mm]
\partial_t m_w + \nabla \cdot J_w = \rho_i R_m W_{\mathrm{SSA}}\left(T_{\mathrm{int}} - T_0\right)
\end{cases}
\tag{A1}
$$

where $J_{u,i}$, $J_{u,w}$, and $J_w$ are heat conduction fluxes in the ice, in the water, and the water flux, respectively, $R_m = \frac{c_w}{\beta L_{\mathrm{fus}}}$ (with $\beta$ the kinetic attachment coefficient for ice growth from liquid water and $c_w$ the thermal capacity of liquid water), $W_{\mathrm{SSA}}$ is the wet surface area, $T_{\mathrm{int}}$ the temperature ice-liquid water interface, $\lambda_i$ and $\lambda_w$ are the ice and liquid water thermal conductivities, and $r_i$ and $r_w$ two microstructural parameters estimated to be $0.06\,d_i$ and $0.081\,d_i$, respectively (with $d_i$ the diameter of the grain in the snow microstructure; Moure et al., 2023). The temperature $T_{\mathrm{int}}$ is given as a linear combination of

the ice, water, and melting temperature:

$$
T_{\mathrm{int}} = \frac{\frac{c_w \rho_w}{\beta} T_0 + \frac{\lambda_i}{r_i} T_i + \frac{\lambda_w}{r_w} T_w}{\frac{c_w \rho_w}{\beta} + \frac{\lambda_i}{r_i} + \frac{\lambda_w}{r_w}} = b_0 T_0 + b_i T_i + b_w T_w.
\tag{A2}
$$

In this system of equations, the temperature of the ice-liquid interface governs the chemical equilibrium. Melting occurs if $T_{\mathrm{int}} > T_0$, freezing occurs if $T_{\mathrm{int}} < T_0$, and local equilibrium is achieved when $T_{\mathrm{int}} = T_0$.





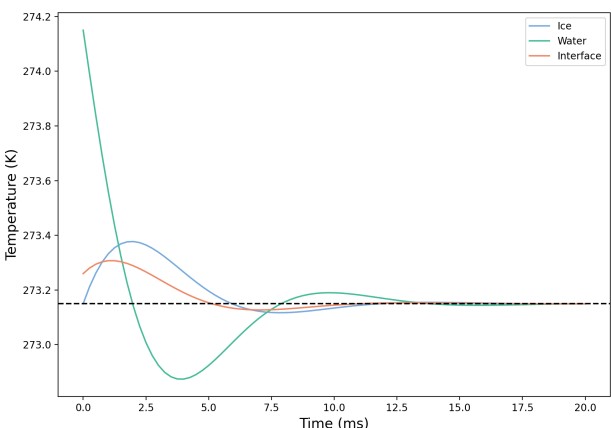

**Figure A1.** Evolution of ice, liquid water, and interface temperatures towards equilibrium, starting from a non-equilibrium situation.

A difficulty in this equation is relating $W_{\mathrm{SSA}}$ to the actual specific surface area. Here, we follow Moure et al. (2023) and assume that $W_{\mathrm{SSA}}$ scales linearly with the saturation degree $S$ (i.e. liquid water filling $50\%$ of the porosity would wet $50\%$ of the interfaces). However, we note that as the liquid water is the wetting phase in snow, it preferentially covers the ice surface, and the interface rapidly gets wet when liquid water is present. Therefore, the assumption that the wet SSA scales linearly with the saturation degree is likely an underestimation, which slows down the relaxation towards thermodynamics equilibrium in snow.

To estimate the timescale of relaxation, we have implemented a simple model simulating relaxation towards local equilibrium (ignoring spatial fluxes for simplicity in the system of Eqs. (A1)). Figure A1 shows the evolution of the temperatures of the ice, the liquid water, and the interface, with $T_{\mathrm{i}} = 273.15 = T_0$ K, $T_{\mathrm{w}} = 274.15$ K, $m_{\mathrm{i}} = 275\,\mathrm{kg\,m^{-3}}$, $m_{\mathrm{w}} = 100\,\mathrm{kg\,m^{-3}}$, and a SSA of $2\,\mathrm{m^2\,kg^{-1}}$ as initial conditions. This illustrates that starting from a situation of non-equilibrium, thermodynamical equilibrium is achieved within about $20\,\mathrm{ms}$. As this timescale of relaxation is much shorter than the timescale of evolution of

the snowpack, the assumption of thermodynamical equilibrium between the ice and liquid water appears justified.

**Appendix B: Constitutive laws**

This appendix presents the constitutive laws chosen for the models. For the thermal conductivity (expressed in $\mathrm{W\,K^{-1}\,m^{-1}}$) we have

$$\lambda = 2.5 \times 10^{-6} \phi_{\mathrm{i}} \rho_{\mathrm{i}} - 1.23 \times 10^{-4} \phi_{\mathrm{i}} \rho_{\mathrm{i}} + 0.024 \tag{B1}$$

and for the saturated hydraulic conductivity (expressed in $\mathrm{m\,s^{-1}}$)

$$K^{\mathrm{sat}} = \frac{g\,\rho_{\mathrm{w}}}{\mu_{\mathrm{w}}} 3 r_{\mathrm{es}}^2 \exp\left(-0.0130 \phi_{\mathrm{i}} \rho_{\mathrm{i}}\right) \tag{B2}$$





where $g = 9.81\,\mathrm{m\,s^{-2}}$ is the gravity acceleration, $\rho_{\mathrm{w}}$ is the density of liquid water and $\mu_{\mathrm{w}} = 1.79 \times 10^{-3}\,\mathrm{Pa\,s}$ is the dynamic viscosity of water at $0\,^{\circ}\mathrm{C}$, and $r_{\mathrm{es}} = 3/(\mathrm{SSA}\rho_{\mathrm{i}}\phi_{\mathrm{i}})$ is the equivalent optical radius. For the $\alpha$ and $n$ parameters, defining the van Genuchten WRC, we have

$$\alpha = 4.4 \times 10^6 \left(\frac{\phi_{\mathrm{i}}\rho_{\mathrm{i}}}{2r_{\mathrm{es}}}\right)^{-0.98} \tag{B3}$$

and

$$n = 1 + 2.7 \times 10^{-3} \left(\frac{\phi_{\mathrm{i}}\rho_{\mathrm{i}}}{2r_{\mathrm{es}}}\right)^{0.61}. \tag{B4}$$

Finally, for the compactive viscosity $\eta$ (expressed in $\mathrm{Pa\,s}$) we have

$$\eta = \frac{3.05 \times 10^{-7}}{1 + 60\theta_{\mathrm{w}}} \frac{\phi_{\mathrm{i}}\rho_{\mathrm{i}}}{250} \exp\left(0.1\left(T_0 - T\right) + 0.023\phi_{\mathrm{i}}\rho_{\mathrm{i}}\right). \tag{B5}$$

## Appendix C: Details on the implementation of models without WRC regularization

In this appendix, we briefly present the implemenations of models 4 and 5, which are based on the treatment of Richards' equation in the SNOWPACK (Wever et al., 2014) and Crocus (D'Amboise et al., 2017) models, respectively. Note, that even though our implementations are based on SNOWPACK and Crocus, they cannot be considered as strict copies. Rather, our implementations mimic how the treatment of Richards' equation in SNOWPACK and Crocus could be translated into other models.

### C1  Model 4

The implementation of this model is based on the SNOWPACK implementation (Wever et al., 2014) and its publicly available source code (`master` branch, `f89d8c17` commit; checked on 16/12/2024). As explained in the main part of the manuscript, we follow here a sequential scheme, where the process of heat conduction, surface energy fluxes, and phase changes are first solved (and liquid percolation treated in a second step). This yields a LWC field, potentially with wet zones that flows and dry zones where the WRC is undefined. This issue is circumvented in two steps: (i) a small amount of liquid water is introduced, to avoid fully-dry media and (ii) the residual point of the WRC is lowered-down such that the LWC lies on a defined part of the WRC. Concretely, to solve Richards' equation, at the beginning of each adaptive timestep:

**1-** We compute the residual LWC of the snow material $\theta_{\mathrm{r}} = \max\left(0, \min\left(\theta_{\mathrm{w}} - \theta_{\mathrm{acc}}/10, \max\left(0, \min\left(0.02, \max(\theta_{\mathrm{r}}^{\mathrm{prev}}, 0.75\theta_{\mathrm{w}})\right)\right)\right)\right)$, with $\theta_{\mathrm{w}}$ the LWC at the start of the timestep, $\theta_{\mathrm{r}}^{\mathrm{prev}}$ the residual LWC used in the previous timestep, and $\theta_{\mathrm{acc}}$ the error aimed for on $\theta_{\mathrm{w}}$ during the non-linear iteration. This step defines a residual LWC that is positive, equals the default value of $0.02$ when



the snow LWC is above $0.027$, and tends to $75\%$ of the LWC when it is below $0.027$. This corresponds to the lines 395 to 397 in the vanGenuchten.cc source file of SNOWPACK.

**2**- We compute the matric potential of "dry" layers. For this, we compute the matric potential of layers if they were just above their residual points. We then compute the minimum (maximum in absolute value) of these matric potentials, and define it as the matric potential that "dry" layers should have. This corresponds to the lines 967 to 969 in the ReSolver1d.cc source file of SNOWPACK.

**3**- We compute the minimum required LWCs that need to be present in the snow layers to reach the "dry" matric potential. This corresponds to line 976 to 999 in the ReSolver1d.cc source file of SNOWPACK.

**4**- If needed, we increase the LWC of layers that fall below their minimum required LWCs (computed in step 3 above), and recompute the residual LWC as in step 1. In order to ensure mass conservation, this is done by taking mass from the ice. This corresponds to the lines 1017 to 1024 in the ReSolver1d.cc source file of SNOWPACK.

After this step, we obtain consistent fields of LWC and WRC for the whole snowpack. Richards' equation (liquid water conservation under matric flow) is solved using the matric potential as the primary potential. For consistency with the other toy-models, we use a Newton method for the non-linear iterations. As with the other models, the stopping criterion is based on the error when trying to close the PDE, with an additional criterion on mass conservation (as using the matric potential as the primary variable affects mass conservation Celia et al., 1990).

## C2   Model 5

Model 5 is based on the Crocus implementation of Richards' equation (D'Amboise et al., 2017) and its source code (`damboise_dev` branch, `67bda59d` commit; checked on 16/12/2024). It follows the overall same strategy as model 4. The main difference relates to the definition of the residual LWC. Concretely, at the beginning of each (adaptive) timestep:

**1**- We compute the residual points of the snow material as $\theta_{\mathrm{r}} = \min\left(0.02, 0.75\theta_{\mathrm{w}}\right)$, with $\theta_{\mathrm{w}}$ the LWC at the start of the timestep. This results in a residual LWC that equals the standard value of $0.02$ in well-wet material ($\theta_{\mathrm{w}} > 0.027$), and that can drop to zero in the case of fully-dry material. This corresponds to the lines 3417 in the snowcro.F90 source file of Crocus.

**2**- We compute the matric potential of "dry" layers. This corresponds to the minimum matric potential of the layers when they are at the so-called "pre-wetting" level ($\theta_{\mathrm{w}} = 10^{-5}$; D'Amboise et al., 2017). This corresponds to the lines 3408 to 3438 in the snowcro.F90 source file of Crocus.

**3**- If the LWC of a cell is below the "pre-wetting" level, the LWC is increased so that the matric potential reach the "dry" matric potential. This is done without ensuring mass conservation.

**4**- The residual points are re-computed as in step 1.

After this step, liquid water matric flow is solved using the same strategy as in model 4.



*Code and data availability.* The implementations of the five toy models have been published as Fourteau (2025), available at
https://doi.org/10.5281/zenodo.14753491.

*Author contributions.* The research was designed by KF, JB, and MD. The mathematical formulation was derived by KF, JB, CC, and MF.
The code was developed by KF with help from CC. The manuscript was written by KF with the help of all co-authors.

*Competing interests.* The authors declare having no competing interests.

*Acknowledgements.* We acknowledge the SNOWPACK and Crocus developers, notably for making their source code easily available.
We thank Laurent Oxarango, Nander Wever and Michael Lombardo for the fruitful discussions on liquid water transport in snow. Kévin
Fourteau's current position and Julien Brondex's past position were funded by the European Research Council (ERC) under the European
Union's Horizon 2020 research and innovation program (IVORI; grant no. 949516). This work was supported by the S-NOW project funded
by the Institut des Mathématiques pour la Planète Terre.



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
