# Peer review of "Numerical strategies for representing Richards' equation and its couplings in snowpack models"

_EGUsphere, 2025_

## Author Comment (AC1)

**Response to Review 1 of egusphere-2025-444**

We are thankfull Richard Essery for taking the time to review our manuscript and for his constructive review. Please find below our point by point response to the review. The comment of the referee are shown in blue and our response in black below. Proposed modifications of the manuscript are shown in green with page and line numbering corresponding to the preprint version of the article.

This is a good paper, and I suggest that it will be publishable with corrections that are merely clarifications or editorial.
1 Vapour transport can also be important; it is only in line 79 that we learn that it is neglected here.

We propose to explicitly say at the beginning of Section 2 that we focus our analysis on liquid water percolation and thus neglect some of the physical processes at play in snowpacks, such as metamorphism or water vapor transport. We will remove the mention to water vapor **L79** concerning the energy equation, and we will add at the start of Section 2 **L72**:

"As we focus this article on liquid water percolation, we neglect several processes at play in snowpacks, such as metamorphism or water vapor transport, in order to simplify our analysis."

2 The highly relevant paper by Wever et al. (2014) is almost "in the last decade", but modelling percolation of water in snow under gravity and capillarity goes back at least as far as Colbeck (1974). https://doi.org/10.3189/S002214300002339X
27 Between bucket schemes and solving the Richards equation, an intermediate approach of calculating water percolation under gravity without capillarity is used in some models (e.g., SNTHERM).

Following the comments from this review and that of the second referee, we propose to rewrite this section of the introduction to better describe the different representation of liquid water flow that have been proposed in models, i.e. with or without capillary pressure gradient, with or without preferential flow, and in 1D or multi-D. We will also specify at the end of the introduction that we restrict our study to the case without preferential flow. We propose to rewrite the paragraph starting **L27** to:

"A simple and largely employed way of representing liquid water percolation in 1D snowpack models is the so-called bucket-scheme. In this picture, snow layers are expected to retain liquid water until a certain threshold, after which all liquid water is instantaneously transferred downward (Bartelt and Lehning, 2002, Vionnet et al., 2012, Sauter et al., 2020). While this implementation is numerically efficient, it cannot capture certain effects, such as capillary barriers, capillary rise, or the finite dynamics of the percolation process. On the other hand, a more detailed description of liquid water flow in snowpacks can be achieved by explicitly solving the liquid water budget under gravitational and capillary forces, i.e. Richards' equation (Richards, 1931capillary, Colbeck, 1974, Illangasekare et al., 1990, Daanen and Nieber, 2009). Richards' equation has notably been implemented in the detailed 1D models SNOWPACK (Wever et al., 2014) and Crocus (d'Amboise et al., 2017). This more advanced description has notably been shown to better capture the timing associated with the wetting of the snowpack (Wever et al., 2015). In the case of significantly wet snow, capillary forces become negligible and the driving force of liquid water flow reduces to gravity only (Colbeck, 1972). This offers a simplified version of Richards' equation, for instance implemented in the SNTHERM 1D model(Jordan, 1991). However, note that implementations based on the standard 1D

Richards' equation cannot represent preferential flow, which is crucial to fully capture the complexity of liquid water percolation in snowpacks (Marsh and Woo, 1985, Schneebeli, 1995, Waldner et al., 2004). Explicit representations of preferential flow in snow have been proposed in the literature. A first broad class of strategies is based on modelling the snowpack in multi-dimensions, allowing the formation of fingering flows in response to snow heterogeneities (Hirashima et al., 2014, Leroux et al., 2017, Leroux et al., 2020) and/or instabilities in the wetting front (Moure et al., 2023). These studies offer valuable insights on the physical mechanisms responsible for the formation of preferential flow in snow. A second strategy, which is compatible with a 1D framework, is the use of a dual-domain percolation model (Wever te al., 2016, Queno et al., 2020)."

74 Is this transposition to 2D or 3D arrays of 1D columns? Inclusion of lateral flows is not so straightforward.
We did not have an array 1D columns in mind to construct a 2D or 3D version, but really the writing of Richards' equation in a multi-dimensional setting, as done for instance in rock sciences. The translation into the general 3D case requires to change the gravity term *cos γ* to the vertical vector z in Richards equation and to potentially change the scalar conductivities into tensor in the case of an anisotropic material. Lateral flow is handled based on the gradient of water potential, similarly as to vertical flow. But this picture requires to treat the whole 2D or 3D at once, rather than splitting it connected 1D columns (spatial decomposition of the domain is always possible, but requires some additional techniques not discussed in the manuscript). This will be precised in the manuscript **L73**

"Note that while this article assumes a 1D framework, as usually done in snowpack models, it could be transposed to a 2D or 3D configuration similar to what is done in several rock, soil, or even some snospack models (Vauclin et al., 1979, Hirashima et al, 2014, Leroux et al., 2017,  Cockett et al., 2018). Therefore, we tried to keep the notation used in this article as general as possible."

We will also add L**127**
"Note that in the multidimensional case, the gravity term cos γ should be replaced by the unit vector orientated with gravity"

81 If wanting to retain *F*cond as a vector for generality, GMD guidelines require it to be printed in boldface. Alternatively, as it is a scalar in the 1D framework, the divergence could simply be $\partial_z F$cond.
We will write $\mathbf{F}_{cond}$ (and other vectors) in boldface to make it explicitly a vector. On a similar note, we will also modify the writing convention of the fluxes in Appendix A from "J" to "**F**" so it is consistent with the main part of the article.

Figure 1 Does the inset serve any useful purpose? Mention it in the caption if so, and remove it if not.
The goal of the insert was to have a better view of the regularization and of the plateau of the WRC. We will mention it in the caption:
"Examples of the regularized water retention curves used in this work, for three different snow density and surface specific area (SSA). The zoomed insert focuses on the regularization and the associated plateau below the retention point."

234 There are models that allow liquid water in snow below the fusion temperature, e.g., https://doi.org/10.1175/2010JHM1249.1 https://doi.org/10.1002/2016WR019672
We will mention in the manuscript that some models assume that the ice/liquid water translation in snowpack occur on a temperature range rather than at a single temperature.

This will be inserted in Section 2.4, where we will rewrite the discussion on the assumption of thermodynamical equilibrium between the ice and the liquid water. We will rephrase the manuscript to

"As most snowpack models (e.g., Jordan, 1991, Bartelt and Lehning, 2002, Vionnet et al., 2012detailed, Sauter et al. 2020) we assume (i) that liquid water and the snow are in thermodynamical equilibrium (which means that the melting/freezing dynamics can be assumed as infinitely fast) and (ii) that this equilibrium occurs at the single temperature $T_0$. However, we note that these assumptions are not systematic in snowpack models. […] Also, due to capillary effects, the thermal equilibrium between the ice and liquid water phases technically occurs on a temperature range rather than at a single temperature. This effect is commonly taken into account in soil models through a so-called soil Freezing Characteristic Curve (soil FCC; Devoie et al. 2022). Some snowpack models have proposed to introduce a similar FCC for snow (Daanen and Nieber 2009, Dutra et al., 2010, Clark et al., 2017). While the FCC of snow could in principle be computed from the WRC of Sect. 2.2 (as done for instance in Daanen and Nieber, 2009, Li et al., 2023), this would represent a significant increase in the complexity of the snow representation. Indeed, the simple equilibrium condition that ice and liquid water can only coexist at $T_0$ would have to be replaced by an implicit equation relating the temperature to the matric potential. However, we note that the computation of a FCC from a diverging WRC implies that thermodynamical equilibrium cannot be reached with a LWC below the divergence point (Daanen and Nieber, 2009), and thus that regularizing the WRC is a necessary step to model a dry material."

326 The harmonic average seems to be the natural choice, corresponding to adding the conductances in adjacent layers in series.
This indeed amounts to having the conductances in series. This will be mentioned in the manuscript **L326**
"This is consistent with the idea that the conductances corresponding to adjacent cells are placed in series. It notably ensures that the heat flux vanishes when the thermal conductivity of one of the two cells vanishes (Kadioglu et al., 2008)."

344 Models 4 and 5 have not yet been introduced.
We propose to rephrase **L344** to
"In two of the implementations that will be presented below (denoted models 4 and 5), this criterion is complemented with a criterion on mass conservation, as these numerical scheme are not naturally mass-conservative."

Also we will remove the mention to models 4 and 5 at the end of the paragraph and rephrase **L361** to:

"As a test, we also run some simulations using the modified Picard rather than the Newton method."

397 It would be good to show temperature, density and SSA for this stratigraphy.
We will add a figure presenting the initial state of the snowpack in terms of density, SSA, and temperature introduced **L425**

"The initial conditions for the density, SSA, and temperature are displayed in Figure 2."

with caption

"Initial conditions of the snowpack used in the simulation, in terms of density (panel a), SSA (panel b), and temperatures (panel c). Note that the initial temperatures are lower in test cases 1 and 3 in order to simulate liquid water infiltration withing a colder snowpack."

Something that we forgot to mention in the first version of the manuscript is that the test cases have the same initial density and SSA, but different initial temperature field. The idea was to perform simulations with more or less cold snowpack (and thus with potentially more or less deep refreezing). This will be mentioned **L398**
"For the initial state of the simulation, the initial temperature was decreased compared to the Crocus output in order to obtain a cold and dry snowpack near its peak snow water equivalent.",

**L413**

"The initialization is the same as in case 1, based on the output of the same Crocus simulation, but with a higher temperature in order to have a snowpack close to its melting point."

and **L420**

"The initialization is also based on a Crocus simulation, with a temperature field intermediate between test case 2 and 3."

Figure Where is the water that appears at the base of the snow before the surface melt water arrives coming from?
The formation of liquid water at the base results from the heat flux used as a bottom boundary condition meant to emulate a warm ground. In the case of a zero heat flux condition this melting is not present. This will be mentioned in the manuscript **L426** alongside a more detailed description of Figure 3:

"In these three cases, liquid water is produced directly at the bottom of the snowpack in response to the 10 W m-2 heat flux from the warm ground. In the absence of such heat flux, the bottom of the snowpack would remain dry until liquid water percolates through the whole snowpack."

Figure 4 Why do increasing timesteps run right to left on the x axis? – not wrong, but unconventional if there is not a clear reason.
There was no specific reason for that choice. We will redo the figure with increasing timesteps running left to right. Note that there are two other modifications to the figure:
- As noted by the second referee, rain was missing from the forcing of Experience 2. This was changed and the figure updated with the new results.
- Moreover, we were not able to reproduce the results for model 5 in Experiences 1 and 3, even after re-running all the simulations (the results for the other models were reproduced). We do not know what was the error in the results of model 5 we used for the initial submission. However, it does not change the conclusion of the manuscript.

505 "internal ice layers" sounds like horizontal layers are being discussed, whereas I think it is actually vertical columns.
In this part we wanted to mention the idea that internal horizontal ice layers can be formed by the injection of liquid water through percolation chimneys down to a capillary barrier where the water can then horizontally spread. When excavated after refreezing, there indeed is a structure composed of vertical ice columns connecting horizontal crusts (as

illustrated by picture #58 of Fierz et al., 2009). We propose to clarify this point by rephrasing **L503** to

"While the exact mechanisms, and therefore description, of preferential flow in snowpacks remain unclear at this point (Hirashima et al., 2014, Hirashima et al., 2019, Moure et al., 2023), the presence of fast flowing, and out-of-equilibrium with the rest of the snow layer, liquid water appears as a prerequisite for the formation of internal horizontal ice layers. In this picture, the preferential flow transports liquid water through cold snow layers down to a capillary barrier, where the liquid water can then horizontally spread and refreeze as a horizontal crust (Queno et al., 2020). This is illustrated by the close relation between refrozen preferential paths (i.e. ice columns) and internal horizontal crusts, for instance illustrated in picture #58 of Fierz et al. (2009)."

Minor corrections:
Below are a few responses to some specific comments. For the rest of the comments, the modifications proposed by Richard Essery will be directly followed.

The text uses both "Richard's equation" and "Richards' equation", and it is often "the Richards equation" in literature. Pick one!
We will make the naming consistent throughout the manuscript as "Richards equation".

22 I'm not sure of the authors' intended emphasis, but "likely" is not the right word here.
We will rephrase "likely" by "similarly".

336 $1/\cos \gamma$
We will remove the parentheses in Eq. 19 and will add the forgotten *cos* **L336**. Unless we made a mistake computing the vertical projection of a length d perpendicular to the slope, we believe it is cos γ d.

457 "can be efficiently cheapened" (or "can be made more efficient" would be better)
We have replaced the end of the sentence with "can be made more efficient".

**REFERENCES**
Cockett, R., Heagy, L. J., and Haber, E.: Efficient 3D inversions using the Richards equation, Computers Geosciences, 116, 91–102, https://doi.org/https://doi.org/10.1016/j.cageo.2018.04.006, 2018

Colbeck, S. C.: A Theory of Water Percolation in Snow, J. Glaciol., 11, 369–385, https://doi.org/10.3189/S0022143000022346, 1972.

Colbeck, S. C.: The capillary effects on water percolation in homogeneous snow, J. Glaciol., 13, 85–97, https://doi.org/10.3189/S002214300002339X, 1974.

Daanen, R. P. and Nieber, J. L.: Model for Coupled Liquid Water Flow and Heat Transport with Phase Change in a Snowpack, J. Cold Reg.760 Engin., 23, 43–68, https://doi.org/10.1061/(ASCE)0887-381X(2009)23:2(43), 2009

Devoie, E. G., Gruber, S., and McKenzie, J. M.: A repository of measured soil freezing characteristic curves: 1921 to 2021, Earth System Science Data, 14, 3365–3377, https://doi.org/10.5194/essd-14-3365-2022, 2022.

Dutra, E., Balsamo, G., Viterbo, P., Miranda, P. M. A., Beljaars, A., Schär, C., and Elder, K.: An Improved Snow Scheme for the ECMWF Land Surface Model: Description and Offline

Validation, Journal of Hydrometeorology, 11, 899 – 916, https://doi.org/10.1175/2010JHM1249.1, 2010

Hirashima, H., Yamaguchi, S., and Katsushima, T.: A multi-dimensional water transport model to reproduce preferential flow in the snowpack, Cold Reg. Sci. Tech., 108, 80–90, https://doi.org/10.1016/j.coldregions.2014.09.004, 2014

Hirashima, H., Avanzi, F., and Wever, N.: Wet-Snow Metamorphism Drives the Transition From Preferential to Matrix Flow in Snow, Geophysical Research Letters, 46, 14 548–14 557, https://doi.org/10.1029/2019GL084152, 2019

Illangasekare, T. H., Walter Jr., R. J., Meier, M. F., and Pfeffer, W. T.: Modeling of meltwater infiltration in subfreezing snow, Water Resources Res., 26, 1001–1012, https://doi.org/10.1029/WR026i005p01001, 1990.

Leroux, N. R. and Pomeroy, J. W.: Modelling capillary hysteresis effects on preferential flow through melting and cold layered snowpacks, Advances in Water Resources, 107, 250–264, https://doi.org/10.1016/j.advwatres.2017.06.024, 2017.

Leroux, N. R., Marsh, C. B., and Pomeroy, J. W.: Simulation of Preferential Flow in Snow With a 2-D Non-Equilibrium Richards Model and Evaluation Against Laboratory Data, Water Resources Research, 56, e2020WR027 466, https://doi.org/10.1029/2020WR027466, 2020

Li, X., Zheng, S.-F., Wang, M., and Liu, A.-Q.: The prediction of the soil freezing characteristic curve using the soil water characteristic curve, Cold Regions Science and Technology, 212, 103 880, https://doi.org/10.1016/j.coldregions.2023.103880, 2023.

Marsh, P. and Woo, M.-K.: Meltwater Movement in Natural Heterogeneous Snow Covers, Water Resources Research, 21, 1710–1716, https://doi.org/10.1029/WR021i011p01710, 1985

Schneebeli, M.: Development and stability of preferential flow paths in a layered snowpack, IAHS Publications-Series of Proceedings and Reports-Intern Assoc Hydrological Sciences, 228, 89–96, 1995

Waldner, P. A., Schneebeli, M., Schultze-Zimmermann, U., and Flühler, H.: Effect of snow structure on water flow and solute transport, Hydrological Processes, 18, 1271–1290, https://doi.org/10.1002/hyp.1401, 2004

---

## Author Comment (AC2)

**Response to Review 2 of egusphere-2025-444**

We are thankfull to the referee for their time reviewing our manuscript and for their helpful remarks. Please find below our point by point response to the review. The comment of the referee are shown in blue and our response in black below. Proposed modifications of the manuscript are shown in green with page and line numbering corresponding to the preprint version of the article.

This paper is of high quality, discussing various implementation strategies for Richards equation in snow cover models. It focusses on an improved treatment of the dry limit of the equation, as well as on a tight coupling with phase changes. This indeed addresses one of the outstanding questions in the field of snow modeling. Some simple test cases are run, to demonstrate the various approaches and setups. The writing is mostly of high quality. I generally think that the paper can be published, after taking my minor considerations into account.

I have a few issues with somewhat more important feedback.

1. I would like to see the discussion of existing literature improved. It is too focused on the recent studies, and too focused on the SNOWPACK and CROCUS models. Examples:

- L31: "it has been proposed in the last decade": I think the earliest implementation of Richards equation in a snow model I am aware of is Jordan 1983 (https://agupubs.onlinelibrary.wiley.com/doi/abs/10.1029/WR019i004p00979), but there is also work by Colbeck, Illangasekare et al., 1990. I think Daanen and Nieber (2009) were also using Richards equation in their model, before SNOWPACK. I think the paper should provide a bit more historical context, even though there is not a need to do an extensive literature review. It's more that I think it's important to provide a proper historical perspective and context.
- The sentence in L61 "Rather, snowpack models rely on a ... 1D framework" is too generalized and not acknowledging the work that has been done extending modeling to 2D and 3D. For example the work by researchers Webb, Leroux, Hirashima. I would like to see the work of those researchers, and maybe others I overlooked, included in the discussion.

Following the remarks of the two referees we propose to modify the introduction to better discuss articles using Richards' equation (with or without a matric potential gradient) before their implementation into SNOWPACK or Crocus. We will also mention the simulations performed in multi-dimensions aimed at explicitly representing preferential flow in snow. We propose to rewrite the paragraph starting **L27** to

“A simple and largely employed way of representing liquid water percolation in 1D snowpack models is the so-called bucket-scheme. In this picture, snow layers are expected to retain liquid water until a certain threshold, after which all liquid water is instantaneously transferred downward (Bartelt and Lehning, 2002, Vionnet et al., 2012, Sauter et al., 2020). While this implementation is numerically efficient, it cannot capture certain effects, such as capillary barriers, capillary rise, or the finite dynamics of the percolation process. On the other hand, a more detailed description of liquid water flow in snowpacks can be achieved by explicitly solving the liquid water budget under gravitational and capillary forces, i.e. Richards' equation (Richards, 1931capillary, Colbeck, 1974, Illangasekare et al., 1990, Daanen and Nieber, 2009). Richards' equation has notably been implemented in the detailed 1D models SNOWPACK (Wever et al., 2014) and Crocus (d'Amboise et al., 2017). This more advanced description has notably been shown to better capture the timing associated with the wetting of the snowpack (Wever et al., 2015). In the case of significantly wet snow, capillary forces become negligible and the driving force of liquid water flow reduces to gravity only (Colbeck, 1972). This offers a

simplified version of Richards' equation, for instance implemented in the SNTHERM 1D model(Jordan, 1991). However, note that implementations based on the standard 1D Richards' equation cannot represent preferential flow, which is crucial to fully capture the complexity of liquid water percolation in snowpacks (Marsh and Woo, 1985, Schneebeli, 1995, Waldner et al., 2004). Explicit representations of preferential flow in snow have been proposed in the literature. A first broad class of strategies is based on modelling the snowpack in multi-dimensions, allowing the formation of fingering flows in response to snow heterogeneities (Hirashima et al., 2014, Leroux et al., 2017, Leroux et al., 2020) and/or instabilities in the wetting front (Moure et al., 2023). These studies offer valuable insights on the physical mechanisms responsible for the formation of preferential flow in snow. A second strategy, which is compatible with a 1D framework, is the use of a dual-domain percolation model (Wever te al., 2016, Queno et al., 2020)."

2. Section 2.2.1: I think it is critical to discuss that this discrepancy in the dry limit is partly because drying water retention curves are being used (i.e., water retention curves derived from drying snow samples). In a wetting snow sample, the water retention curve may actually nicely approach 0 residual water content. Hysteresis has been used in snow modeling in other studies: Leroux and Pomeroy (2017): https://www.sciencedirect.com/science/article/pii/S0309170817300040.
To the best of our understanding, it is true that WRCs that has been obtained under draining condition tend to diverge at a finite LWC while imbibition curves can be better-behaved and reach a vanishing LWC. This is notably visible in recent numerical experiments of Bouvet et al. (preprint; EGUsphere). If we are not wrong this comes from the fact when draining, the liquid phase can become isolated and suction applied at the boundary of the sample therefore cannot moved the water inside. We will mention this point in the manuscript. However, we think that the use of hysteresis does not fully solve the problem. First, some hysteresis model predicts a divergence of the WRC at a finite LWC both for the drying and wetting curves (for instance in Leroux and Pomeroy, 2017). Secondly, the divergence of the draining WRC remains a problem, as it would suggest that once wet snow cannot be dried again. We propose to add **L159**:

"The use of a WRC with a hysteresis distinguishing the drying and wetting curves (for instance as in Leroux and Pomeroy, 2017) could partially mitigate this problem, as wetting curves can reach a vanishing LWC (Bouvet et al., preprint). However, even in this case, the issue of a diverging WRC at a finite LWC remains a problem for drying snow."

3. I think that the paper currently insufficiently discusses that implicit methods can be numerically stable, but still that doesn't mean that they are accurate. I think this is important to convey to the reader. So for example, I would not write in the conclusions: "which can run with timesteps of 15 min and above", without immediately making a remark about the accuracy of the solution. The reason I'm saying this, is because in Fig. 4, it looks like that the most advanced models perform worse at the largest time steps. That is just something to be very careful about. I also wonder if it is not better to show two Figures for Fig. 4: one where each numerical approach is tested for time step sensitivity, by taking 1s simulations for each of the approaches as a benchmark. And the other the current Fig. 4, showing the estimated accuracy under the assumption that Model 1 at 1s timesteps can serve as a benchmark. That would also more clearly deal with the fact that if model 1 run at 1 second time step is considered the reference for determining error-statistics, like RMSE, it is quite obvious that model 1 performs best and creeps closer and closer to the reference simulation when the time step reduces. I wonder if Models 3-5 actually just have another solution, and that for that reason, their performance is relatively constant, compared to the benchmark.

Indeed, the robustness and accuracy of model are two relatively independent metrics. What we wanted to mean by "which can run with timesteps of 15 min and above" and that some models are actually able to solve Richards' equation with such a timestep, without the need to internally drop the timestep. Of course, increasing the timestep results in a degradation of the accuracy. This will be mentioned in the manuscript **L539**:

"which can run with timesteps of 15 min and above without the need for a shorter internal timestep (although using large timesteps naturally results in a degradation of the simulations' accuracy), while the non-regularized models require shorter timesteps to handle liquid water percolation."

We will also specify **L426** that while choosing model 1 is not neutral. That being said it can still be motivated, notably because operator-splitting is known to introduce errors (Connors et al., 2014):

"Finally, we note that using model 1 as a benchmark is not neutral, as it places this model in a specific position compared to the others. This choice was made as we expect its physics to be the most cleanly defined (without diverging WRC and artificial displacement of the residual LWC to circumvent this divergence) and that the use of operator-splitting is known to introduce errors (Keyes et al. 2013, Connors et al., 2014)."

For figure 4, we think it is important to keep in mind that models 4 and 5 (and 3 to some degrees) require the use of an adaptative timestep that drops below 15min. So while the default timestep size is large, the resolution of the liquid water percolation actually occurs at a short timestep. This gives them an advantage compared to the models that stick to the default timestep size. We will mention this point in the manuscript **L465**:

"Note that because of the presence of an adaptive timestep for solving liquid water percolation, the value of the default timestep size should be interpreted with caution in Figure 4. Indeed, as models 4 and 5 (and 3 to some degree) require small adaptive timesteps, their solving of liquid water percolation actually occurs with a timestep much lower than the default value. This gives them an advantage in terms of RMSD, as errors stemming from the temporal discretization of Richards' equation are reduced in the process."

Finally, we have run the other models at 1s to see their behavior with vanishing timesteps. What we found is that:
- Models 1 and 2 converge to the same solution (and model 1 converges a bit faster as illustrated in Figure 4).
- Models 4 and 5 do not converge to this solution.
- Model 3 does not converge to this solution neither.

While it could be expected that models 4 and 5 do not converge to the same solution as 1 and 2 since they have a slightly different physics for their WRC, the fact that model 3 do not agree with 1 and 2 is more surprising as they in principle share the same physics. Also it is unclear that model 3, 4, and 5 actually converge when the timestep is reduced. Below are the output of model 3 in test case 3 have performed simulation with timesteps 112s, 1s, and 0.25s. While the solution at 112s is very close to that of the benchmark with model 1, model 3 gets further away from it every time the timestep is reduced. Notably a drop of 0.75s between the 1s and 0.25s produces a significantly different results. Unfortunately, the numerical cost of running the models at even smaller timestep is to high to see if the model eventually converges to a solution with smaller timesteps.

[Figure]

*Model 3 112s*    *Model 3 1s*    *Model 3 0.25s*

After investigation it seems that this behavior is related to the presence of phase changes. We performed simulations with phase changes (either by having the snowpack at the fusion point with rain input, or by having a cold snowpack with radiation input), and model 3 because equivalent to model 1 in this case (as expected as they now represent the same physics). But we were not able to further pin-point the origin of this behavior.

This will be mentioned in the manuscript **L468**:

"As expected, the overall trend is a decrease of the RMSDs with small timesteps. However, while the unregularized models 4 and 5 perform similarly as the other models with large default timesteps, they do not show a clear decrease of error at smaller timesteps. Our understanding is that the strategy of modifying the residual LWC based at the start of each timestep impacts the physics and changes the solution to which the models converge with small timesteps. This results in a plateau or even an increase of the associated RMSDs in Fig.4. However, it further appears that model 3 does not show a clear convergence for vanishing timesteps as well, despite having in principle the same physics as model 2 and 3. This is visible in the tendency of the RMSD model 3 to plateau for timesteps below 225s. After investigation, we found that model 3 actually diverges away from the benchmark solution for very small timesteps (of the order of the second). The same behavior for very small timesteps was seen for models 4 and 5 as well. This is puzzling as it suggests that either  (i) models 3, 4, and 5 require timesteps well below 1s to reach convergence, or that (ii) they do not have a converging solution with vanishing timesteps. Unfortunately, due to high numerical cost, we were not able to run the models well below the 1s timestep to further explore this behavior."

We will also mention this point in Section 5.3 **L490 :**

"However, as explained in Sect. 5.2, it appears that the models with phase changes decoupled from heat conduction and liquid percolation (i.e. models 3, 4, and 5) do not yield well-converged solution with timesteps of 1 s. While we were not able to precisely explain this behavior, the fact that it appears in models 3, 4 and, 5 might suggest that it is linked with the use of operator-splitting for phase changes. However, this point needs to be further investigated. Therefore, the standard practice of operator-splitting seems to be overall well-justified, at least with timesteps of the order of 900 s and when representing

the interaction of matric flow with heat conduction and phase changes in snowpack models."

I also have a few minor comments:

L68: "effective" may require a definition in this context.
We will specify why macroscopic properties in porous medium are called effective **L68**:

"In this framework, snow is characterized by macroscopic material properties (sometimes referred to as effective properties, as they characterize the behavior of the porous medium treated as an equivalent homogeneous material; Auriault et al., 2009)"

L79: "neglecting the influence of water vapor": isn't the effect of water vapour included in lambda, as is common for snowcover models?
Taking into account the effect of water vapor in the snow requires to modify both the energy conservation and mass conservation equations. In the case of energy, this can indeed be done by increasing the thermal conductivity under the assumption that water vapor is highly reactive with the ice matrix, but this approach does not directly work in the case of a less reactive vapor (Calonne et al., 2014). Also taking into water vapor movement in the presence of liquid water remains an open question. Thus, we have decided to neglect water vapor transport all together in this article. This should have been mentioned as an overall simplifying assumption at the start of Section 2 rather than mentioned in the case of the energy equation only. In line with a comment from Referee 1, we propose to remove the mention to water vapor **L79** and to mention it with other simplifying assumptions **L72**:

"As we focus this article on liquid water percolation, we neglect several processes at play in snowpacks, such as metamorphism or water vapor transport, in order to simplify our analysis."

Section 2.1: I found this section somewhat overly complex. I don't fully understand why the possibility of phase changes is not directly included in Eq. 1. Now, Eq. 5 is basically incompatible with Eq. 1, because melt is introduced at a later stage. I think this section could be simplified in this regard.
Our idea was to start from the homogenized equation that has been derived in Calonne et al., (2014) and to add the missing ingredient of melt/refreeze. In retrospect, we agree that this overly complexified the presentation. Following this comment, we will reshape the section to directly present Eq.(5) with a source term due to phase change.
"As a first equation governing the evolution of a snowpack, we consider the energy conservation of snow, understood here as the combination of the ice matrix and the air within (and excluding potential liquid water). The temporal evolution of the snow energy is given by a classical conservation equation, i.e.

[ENERGY BUDGET OF SNOW WITH PHASE CHANGE TERM]

where $h_s$ is the energy content of snow (expressed in J m$^{-3}$, $F_{cond}$ the heat conduction flux (in W m$^{-2}$), $Q_{abs}$ a volumetric energy source (in W m$^{-3}$) due to shortwave absorption within the snowpack (Van Dalum et al., 2019, Picard et al., 2024), and $Q_{freeze}$ the energy absorbed/released during freezing/melting (in W m$^{-3}$)."

Also to streamline the presentation of the equation, the discussion on the thermodynamic equilibrium condition between the ice and liquid water phase has been entirely moved to Section 2.4.

"In order to close the system of equations, it is necessary to provide an expression for the freezing rate Mfreeze. This is done by defining both the thermodynamical equilibrium between ice and liquid water and the dynamics with which this equilibrium is restored. However, to the best of our knowledge, there is no broadly accepted theoretical or experimental work providing the dynamics of melting or freezing of water in snow (Moure et al., 2023). As most snowpack models (e.g., Jordan, 1991, Bartelt and Lehning, 2002, Vionnet et al., 2012, Sauter et al. 2020) we assume (i) that liquid water and the snow are in thermodynamical equilibrium (which means that the melting/freezing dynamics can be assumed as infinitely fast) and (ii) that this equilibrium occurs at the single temperature $T_0$. However, we note that these assumptions are not systematic in snowpack models. First, the recent article of Moure et al. (2023) proposes to relax the assumption of local thermal equilibrium and to introduce a finite rate of phase change, derived from the upscaling of the Frenkel-Wilson equation. This implies that the ice and liquid water temperatures in a snow sample are in general different and can be above or below the fusion point $T_0$. This modeling framework, composed of four partial differential equations, is briefly presented in Appendix A. However, as observed in the Appendix, the timescale of relaxation towards local thermodynamical equilibrium appears to be much smaller than the timescale of matric water movement and heat diffusion considered in this manuscript, which supports the standard assumption of local equilibrium in snowpack models. Also, due to capillary effects, the thermal equilibrium between the ice and liquid water phases technically occurs on a temperature range rather than at a single temperature. This effect is commonly taken into account in soil models through a so-called soil Freezing Characteristic Curve (soil FCC; Devoie et al. 2022). Some snowpack models have proposed to introduce a similar FCC for snow (Daanen and Nieber 2009, Dutra et al., 2010, Clark et al., 2017). While the FCC of snow could in principle be computed from the WRC of Sect. 2.2 (as done for instance in Daanen and Nieber, 2009model, Li et al., 2023), this would represent a significant increase in the complexity of the snow representation. Indeed, the simple equilibrium condition that ice and liquid water can only coexist at T0 would have to be replaced by an implicit equation relating the temperature to the matric potential. However, we note that the computation of a FCC from a diverging WRC implies that thermodynamical equilibrium cannot be reached with a LWC below the divergence point (Daanen and Nieber, 2009), and thus that regularizing the WRC is a necessary step to model a dry material.

Under our assumptions, the total energy content of both the snow and liquid water $h=h_s + \rho_w L_{fus} \theta_w = c_p (T-T_0) + \rho_w L_{fus} \theta_w$, the temperature T, and the LWC $\theta_w$ become directly related: in the case where h is below the fusion value, $\theta_w$ vanishes and $T<T_0$; in the case where h is above the fusion value, $\theta_w$ is proportional to the excess of energy and $T=T_0$. Specifically, the liquid water content and the temperature can be expressed as a function of the total energy

*[EQS FOR T AND θw]*

Thus, the snowpack becomes governed by solely two PDEs: the total energy budget and the total mass budget. These equations can be obtained by combing Eqs. (1), (6), and (11), which yields

*[TOTAL ENERGY AND TOTAL MASS EQS]*

Note that the rate of freezing/melting $M_{freeze}$ has been eliminated from the system of equations. It can still be derived as a diagnostic from the closure of Richards' equation. It physically corresponds to the amount of frozen and melted water required to maintain the local thermodynamic equilibrium."

L162: "as snow below the fusion is considered dry". In fact, an argument that is sometimes used is that snow contains a thin water film, even below zero. And that for that reason, it can be assumed that theta is never truly 0 in snow. I'm not sure what the authors think about this argument.

As far as we understand, there are two main ways to justify the presence of liquid water below 0°C (neglecting the role of salts/impurities that can also change the fusion point of ice).

The drop of liquid water's pressure due to capillarity forces changes the fusion temperature. Just like in soil, snow thus technically does not melt at a single temperature, but rather over a temperature range following a Freezing Characteristic Curve (FCC), a point that we neglect in our study. This FCC is directly related to the WRC (Daanen and Nieber, 2009, Li al., 2023). The asymptotic divergence of the van Genuchten law at 0.02 that is usually chosen in snow models implies that snow with less 0.02 of liquid water cannot exist (or that if it exists it is out of equilibrium and that the ice would melt to restore at least 0.02 of water), which seems in contradiction with observations. Thus, while it is true that a more realistic representation of snow should include a FCC, diverging WRCs at finite LWC lead to rather unrealistic FCC.

A second argument for the presence of liquid water below 0°C could also be the notion of the so-called quasi liquid layer (QLL) or pre-melting layer. From our point of view (which might be wrong), the existence of a QLL does not really corresponds to the existence of the liquid water phase as predicted by the diverging van Genuchten law. First, the QLL usually spans a few molecular layers. Doing some back of the envelop computations with a 10 molecular layers QLL, a surface density of 12 molecules/nm2 (Wei et al., 2001) and a specific surface of 1000 $m^2/m^3$, the resulting LWC is of about $10^{-6}$ $m^3/m^3$, which is far less than the 0.02 usually found in snow sciences.
Following our response to the comment on Section 2.1 (and a comment by Richard Essery's review) we propose to specify in Section 2.4 that liquid water can technically exist below 0°C due to capillary effects:

"Also, due to capillary effects, the thermal equilibrium between the ice and liquid water phases technically occurs on a temperature range rather than at a single temperature. This effect is commonly taken into account in soil models through a so-called soil Freezing Characteristic Curve (soil FCC; Devoie et al. 2022). Some snowpack models have proposed to introduce a similar FCC for snow (Daanen and Nieber 2009, Dutra et al., 2010, Clark et al., 2017). While the FCC of snow could in principle be computed from the WRC of Sect. 2.2 (as done for instance in Daanen and Nieber, 2009model, Li et al., 2023), this would represent a significant increase in the complexity of the snow representation. However, we note that the computation of a FCC from a diverging WRC implies that thermodynamical equilibrium cannot be reached with a LWC below the divergence point (Daanen and Nieber, 2009), and thus that regularizing the WRC is a necessary step to model a dry material."

 If one would set the hydraulic conductivity to 0, it would be possible to suppress any liquid water flow and tiny liquid water amounts. So it is not a given that there is a tendency for percolation, I think.

It is indeed true that the percolation could be stopped by only applying the reduction of the residual LWC to the WRC and not the hydraulic conductivity. This way, the hydraulic conductivity will reach zero when the LWC drops below the residual LWC. We will rephrase **L165** the manuscript to precise this point:

"Furthermore, this technique requires the residual LWC to be artificially modified, in order to remain strictly below the liquid water content at all times. If this modified residual LWC is applied to both the WRC and the hydraulic conductivity, snow containing very little liquid water will tend to percolate, even though the unmodified WRC would rather imply that the liquid water should be held still under capillary forces. Percolation could be stopped by keeping a hydraulic conductivity that vanishes at the residual point, but the physical link between the WRC and the hydraulic conductivity (Mualem, 1976) would then be partially lost."

 "for melting refreezing event in" is a somewhat broken phrasing
Indeed, there are some missing words **L204**:

"There are two potential ways to account for a melting or refreezing event in the ice budget […]"

 This is actually not the case in SNOWPACK. Maybe it was removed at some point, since Bartelt and Lehning 2002 is cited. But I'm quite sure that phase changes translate volumetric contents between the ice and water phase. If models refreeze by increasing volumetric ice content, and melt by reducing element length, repeated melt-freeze cycles basically generate artificial ice layers, because it is an inconsistent approach. The current approach and reasoning in SNOWPACK is exactly what is described in L209-L212.

We are very sorry to have misreported the strategy used in SNOWPACK to treat melting. Our confusion came from reading the SNOWPACK source code, that we have obviously badly interpreted. It is indeed clearly written in Eqs. (46) and (47) of Bartelt and Lehning (2002) that melt and refreezing are treated symmetrically. We will rephrase L205 to indicate that we follow the same strategy as in SNOWPACK, which we also think makes more physical sense:

"There are two potential ways to account for a melting or refreezing event in the ice budget while respecting mass balance: it can be either viewed as a decrease or increase in density at constant volume, or as a loss or gain of volume at constant density. Depending on the snowpack model, these two possibilities have been employed to represent melt. For instance, melt is treated as a decrease of thickness at constant density in the Crocus model (Vionnet et al., 2012) and a decrease of density at constant thickness in SNOWPACK (Bartelt and Lehning, 2002). The justification for the former choice follows the observation that melting snow is usually of a high density, and thus that the melting process should not act as a de-densification mechanism. Yet, as phase changes occur directly within the snow microstructure, at the surface of the porous ice matrix, we rather believe that both melting and refreezing impact the snow density, without direct impact on the thickness, as done in the SNOWPACK model."

 "Replacing Eq. 5 ..." is somewhat broken phrasing.
Following the rewriting of Section 2.4 we propose to write:

"Thus, the snowpack becomes governed by solely two PDEs: the total energy budget and the total mass budget. These equations can be obtained by combing Eqs. (1), (6), and (11), which yields
[TOTAL ENERGY AND TOTAL MASS EQUATIONS]"

L328: "we discretize as well" not sure this is proper phrasing
We will rephrase to

"For the computation of the liquid water flux, the gradient of the matric potential is also computed based on the average values in the cells and the center-to-center distance."

L333: "when on the of the cell" needs rephrasing
Here "on" should have been "one":

"the liquid water flux vanishes when one of the cell is impermeable"

L346-347: "get stuck in cycles". I would say instead of the algorithm diverges, the solution diverges. And instead of get stuck in cycles, I would say that the solution oscillates.
If the referee agrees, we prefer not to use the word solution (which for us corresponds to the final result that satisfies the discretized equations once the algorithm converged) but iterations instead. We propose to rephrase the sentence to:

"In other words, it is possible for the algorithm to produce diverging iterations or even to produce iterations that oscillate without converging to the solution."

L361-362: This sentence is a bit unclear. I assume that this is not shown in the paper, so I would write: "This did not modify the results (not shown)." To signal to the readers that this is not further discussed.
Yes, the results are mentioned but not shown neither further discussed in the paper. We will follow the proposition of the referee:

"As this did not modify the results of the article (not shown), it will not be further discussed."

L372: "evolution of density". When there is melt, the density can change very rapidly. I suggest to write "the timescales for snow compaction".
The decoupling introduced here is between the density evolution due to phase changes and the evolution of the heat and liquid water contents. So the timescale involved in this decoupling is the evolution of the density due to phase change and not only compaction. We propose to rephrase the sentence **L372** to specify that this assumption might be invalid in case of strong melting.

"The motivation behind it is that the timescale for the evolution of the density is usually longer (of the order of a day, unless in case of abrupt melting) than that of the energy and LWC evolution (of the order of an hour or less)."

L397-398: "To better capture the generally steep gradients of density, temperature, ..., near the surface" is more accurate phrasing I think.
We will rephrase **L397** the sentence according the referee proposition:

"This yields a realistic stratigraphy, with thinner cells near the surface in order to better capture the generally steep gradients of density, temperature, and liquid water content near the surface."

L405: "radiation"
This will be corrected.

L426: I think a brief explanation of what can be seen needs to be added. For example that the daily cycles can be seen in panel a, that ponding can be observed, etc. In this light, I was wondering if the liquid water that can be seen at the snow/soil transition is coming from above, or generated from below due to the soil heat flux? Or is it a suction from the soil that is somehow considered?
We will provide a brief description of the main phenomena observed in Figure 2. We propose to add **L426**:

"As seen in the figure, the first test case shows diurnal surface melt with liquid water infiltrating deeper in the snowpack day after day. Note that a significant amount of liquid water is retained around the 20 cm horizon. This coincides with an abrupt drop in the snow SSA, which acts as a capillary barrier. In the second test case, surface melt is less pronounced due to the lower input radiative fluxes. The constant rain input produces a steady percolation of the liquid water that reaches about the middle of the snowpack after a day. In the last test case, the abrupt rain input results in a fast movement of the percolation front through almost the whole snowpack. As in the first test case, a significant amount of liquid water is retained around the 20 cm horizon. In these three cases, liquid water is produced directly at the bottom of the snowpack in response to the 10 W m$^{-2}$ heat flux from the warm ground. In the absence of such heat flux, the bottom of the snowpack would remain dry until liquid water percolates through the whole snowpack."

Fig. 2b: To me, it looks like the pattern of meltwater infiltration is inconsistent with what was written on L416: "a constant rain flux ...lasting the whole simulation". Because around 0.5 day into the simulation, the downward percolation stops. I think with continued rain, this should also continue to percolate further. Maybe this is something the authors can double check, or explain in the manuscript what exactly happens in the simulation.
You are right that rain was missing in the simulation. After checking the input forcing we realized that the rain flux was actually turned to zero for this simulation set up. We are sorry for this mistake, and re-ran the simulations with rain this time.
The results now show water percolating during the whole simulation, in accordance with the constant rain input. The differences between the models remain overall the same, and thus this does not change the results and discussion of the article.

L457: "can be efficiently cheapen" not sure about this phrasing.
Following the remark of Richard Essery, we will rewrite the end of the sentence to "can be made more efficient".

Fig. 3: There is no need for colors in fact, since each panel shows one line. I think it might be easier to interpret the figure when it is black and white, but I leave it up to the authors.
It is true that the color are unnecessary in this figure, as the models are already separated by panels. Our idea was to the use the same color scheme as in Figure 4 to ease the comparison of the figures. We done two new version of the Figure, with and without colored line in the panels (the text on the left have been changed to black to ease reading).

[Figure]

We have a preference for the colored line version, as it seems less cluttered, but we do not have any problem changing it if required.

**REFERENCES**

Auriault, Jean-Louis, B. C. and Geindreau, C.: Homogenization of Coupled Phenomena in Heterogenous Media, John Wiley Sons, Ltd, 111 River Street, Hoboken, NJ 07030, USA, https://doi.org/10.1002/9780470612033, 2009.

Bouvet, L., Allet, N., Calonne, N., Flin, F., and Geindreau, C.: Simulating liquid water distribution at the pore scale in snow: water retention curves and effective transport properties, EGUsphere, 2025, 1–28, https://doi.org/10.5194/egusphere-2025-2903, 2025

Cockett, R., Heagy, L. J., and Haber, E.: Efficient 3D inversions using the Richards equation, Computers Geosciences, 116, 91–102, https://doi.org/https://doi.org/10.1016/j.cageo.2018.04.006, 2018

Colbeck, S. C.: A Theory of Water Percolation in Snow, J. Glaciol., 11, 369–385, https://doi.org/10.3189/S0022143000022346, 1972.

Colbeck, S. C.: The capillary effects on water percolation in homogeneous snow, J. Glaciol., 13, 85–97, https://doi.org/10.3189/S002214300002339X, 1974.

Connors, J. M., Banks, J. W., Hittinger, J. A., and Woodward, C. S.: Quantification of errors for operator-split advection–diffusion calcula-
tions, Computer Methods in Applied Mechanics and Engineering, 272, 181–197, https://doi.org/10.1016/j.cma.2014.01.005, 2014.

Daanen, R. P. and Nieber, J. L.: Model for Coupled Liquid Water Flow and Heat Transport with Phase Change in a Snowpack, J. Cold Reg.760 Engin., 23, 43–68, https://doi.org/10.1061/(ASCE)0887-381X(2009)23:2(43), 2009

Devoie, E. G., Gruber, S., and McKenzie, J. M.: A repository of measured soil freezing characteristic curves: 1921 to 2021, Earth System Science Data, 14, 3365–3377, https://doi.org/10.5194/essd-14-3365-2022, 2022.

Dutra, E., Balsamo, G., Viterbo, P., Miranda, P. M. A., Beljaars, A., Schär, C., and Elder, K.: An Improved Snow Scheme for the ECMWF Land Surface Model: Description and Offline Validation, Journal of Hydrometeorology, 11, 899 – 916, https://doi.org/10.1175/2010JHM1249.1, 2010

Hirashima, H., Yamaguchi, S., and Katsushima, T.: A multi-dimensional water transport model to reproduce preferential flow in the snowpack, Cold Reg. Sci. Tech., 108, 80–90, https://doi.org/10.1016/j.coldregions.2014.09.004, 2014

Hirashima, H., Avanzi, F., and Wever, N.: Wet-Snow Metamorphism Drives the Transition From Preferential to Matrix Flow in Snow, Geophysical Research Letters, 46, 14 548–14 557, https://doi.org/10.1029/2019GL084152, 2019

Illangasekare, T. H., Walter Jr., R. J., Meier, M. F., and Pfeffer, W. T.: Modeling of meltwater infiltration in subfreezing snow, Water Resources
Res., 26, 1001–1012, https://doi.org/10.1029/WR026i005p01001, 1990.

Keyes, D. E., McInnes, L. C., Woodward, C., Gropp, W., Myra, E., Pernice, M., Bell, J., Brown, J., Clo, A., Connors, J., Constantinescu, E., Estep, D., Evans, K., Farhat, C., Hakim, A., Hammond, G., Hansen, G., Hill, J., Isaac, T., Jiao, X., Jordan, K., Kaushik, D., Kaxiras, E., Koniges, A., Lee, K., Lott, A., Lu, Q., Magerlein, J., Maxwell, R., McCourt, M., Mehl, M., Pawlowski, R., Randles, A. P., Reynolds, D., Rivière, B., Rüde, U., Scheibe, T., Shadid, J., Sheehan, B., Shephard, M., Siegel, A., Smith, B., Tang, X., Wilson, C., and Wohlmuth, 31 B.: Multiphysics simulations: Challenges and opportunities, The International Journal of High Performance Computing Applications, 27, 4–83, https://doi.org/10.1177/1094342012468181, 2013

Leroux, N. R. and Pomeroy, J. W.: Modelling capillary hysteresis effects on preferential flow through melting and cold layered snowpacks, Advances in Water Resources, 107, 250–264, https://doi.org/10.1016/j.advwatres.2017.06.024, 2017.

Leroux, N. R., Marsh, C. B., and Pomeroy, J. W.: Simulation of Preferential Flow in Snow With a 2-D Non-Equilibrium Richards Model and Evaluation Against Laboratory Data, Water Resources Research, 56, e2020WR027 466, https://doi.org/10.1029/2020WR027466, 2020

Li, X., Zheng, S.-F., Wang, M., and Liu, A.-Q.: The prediction of the soil freezing characteristic curve using the soil water characteristic curve, Cold Regions Science and Technology, 212, 103 880, https://doi.org/10.1016/j.coldregions.2023.103880, 2023.

Marsh, P. and Woo, M.-K.: Meltwater Movement in Natural Heterogeneous Snow Covers, Water Resources Research, 21, 1710–1716, https://doi.org/10.1029/WR021i011p01710, 1985

Moure, A., Jones, N., Pawlak, J., Meyer, C., and Fu, X.: A Thermodynamic Nonequilibrium Model for Preferential Infiltration and Refreezing of Melt in Snow, Water Resources Research, 59, e2022WR034 035, https://doi.org/https://doi.org/10.1029/2022WR034035, 2023.

Mualem, Y.: A new model for predicting the hydraulic conductivity of unsaturated porous media, Water Resources Research, 12, 513–522, https://doi.org/https://doi.org/10.1029/WR012i003p00513, 1976.

Schneebeli, M.: Development and stability of preferential flow paths in a layered snowpack, IAHS Publications-Series of Proceedings and Reports-Intern Assoc Hydrological Sciences, 228, 89–96, 1995

Waldner, P. A., Schneebeli, M., Schultze-Zimmermann, U., and Flühler, H.: Effect of snow structure on water flow and solute transport, Hydrological Processes, 18, 1271–1290, https://doi.org/10.1002/hyp.1401, 2004

Wei, X., Miranda, P.B., and Shen, Y.R: Surface Vibrational Spectroscopic Study of Surface Melting of Ice, Phys. Rev. Lett., 86, 1554, https://doi.org/10.1103/PhysRevLett.86.1554, 2001